# Development and application of a NP-cELISA for the detection of nucleoprotein antibodies of equine influenza virus

Yan Yang,[1] Kui Guo,[1] Ling Xu,[1] Wei Guo,[1] Mingqi Dong,[2] Wen Liu,[2] Shuaijie Li,[1] Zenan Zhang,[1] Xiaoyu Chu,[1] Yaoxin Wang,[1] Zhenyu Zhang,[1,3] Zhe Hu,[1] Xiaojun Wang[1,4]

**ABSTRACT**   Equine influenza (EI), caused by the equine influenza virus (EIV), is an acute respiratory disease that has become enzootic worldwide, resulting in frequent outbreaks and substantial economic losses within the equine industry. In this study, we developed a competitive enzyme-linked immunosorbent assay (NP-cELISA) for the detection of antibodies against the EIV nucleoprotein (NP). The assay was designed by coating plates with purified monoclonal antibodies (mAbs) against the NP protein, followed by simultaneous incubation of the test serum samples and HRP-NP antigen in a competitive binding reaction. Receiver operating characteristic (ROC) curve analysis demonstrated that the assay achieved 100% sensitivity and specificity. To assess the diagnostic performance of the NP-cELISA, we evaluated 119 clinical samples in parallel using the NP-cELISA, a commercially available competitive ELISA (ID.vet-cELISA), and the hemagglutination inhibition (HI) assay as the reference standard. The results indicated that the NP-cELISA showed an 87.4% concordance rate with the HI assay, outperforming the 78.2% concordance rate observed between the ID.vet-cELISA and the HI test. Additionally, in a serological surveillance study conducted using the developed NP-cELISA in China from 2021 to 2023, equine serum samples showed an average annual seroprevalence of 37.96% for EIV antibodies. In conclusion, the NP-cELISA developed in this study demonstrates significant potential as a reliable and efficient diagnostic tool for the serological detection of EI, with broad applicability in various settings.

**IMPORTANCE**   Equine influenza (EI) is a highly contagious respiratory disease that poses significant economic and health challenges to the global equine industry. Current diagnostic methods, such as hemagglutination inhibition (HI), are accurate but complex and impractical for widespread use, especially in regions like China where commercial kits are unavailable. This study developed a competitive ELISA (cELISA) for detecting EI virus antibodies, offering a simpler, faster, and more cost-effective alternative. The assay demonstrated higher concordance with HI than existing commercial kits and effectively monitored antibody responses in vaccinated horses. Additionally, it enabled the first large-scale serological survey of EI in China, providing critical insights into the virus's prevalence. This advancement supports timely disease detection and control, benefiting veterinary practices and the equine industry worldwide.

**KEYWORDS**   nucleoprotein, monoclonal antibodies, cELISA, detection, epidemiological

I nfluenza A viruses (IAVs) can cause disease in a wide range of hosts, including humans, equines, pigs, canines, felines, birds, bats, and marine mammals (1–6). IAV is an enveloped virus with a segmented genome consisting of eight single-stranded RNA segments. IAVs are classified into subtypes based on the surface proteins hemagglutinin (HA) and neuraminidase (NA) (7). Like other RNA viruses, IAVs evolve rapidly, and mutations in the HA and NA genes can lead to antigenic changes through a process

**Peer Reviewers** Carmen Guzmán-Bracho, Secretaria de Salud de Mexico, Mexico City, Mexico; Mohammed Filali, Universite Hassan II de Casablanca, Casablanca, Morocco

Address correspondence to Xiaojun Wang, wangxiaojun@caas.cn, Zhe Hu, huzher@126.com, or Zhenyu Zhang, zzhang2527@wisc.edu.

Yan Yang, Kui Guo, and Ling Xu contributed equally to this article. Author order was determined by drawing straws.

The authors declare no conflict of interest.

See the funding table on p. 16.

known as antigenic drift (8). Co-infections with different IAVs can result in reassortment due to the segmented nature of the viral genome. Antigenic shift occurs when reassortment involves the genomic segments encoding HA and/or NA. IAVs are believed to have originated in waterbirds, which remain their natural reservoir. The equine influenza virus (EIV) is thought to have emerged from a direct spillover event from birds (9). Equine influenza (EI) is highly contagious and spreads rapidly among susceptible hosts. It affects equines in most countries, with only Iceland and New Zealand remaining free of the disease (10). EI remains one of the most significant respiratory diseases of horses in the 21st century and has been classified as a notifiable animal disease by the World Organization for Animal Health (WOAH).

Although equine influenza (EI) is generally well-tolerated and rarely fatal, it can significantly reduce productive performance in affected animals. Additionally, the EI virus can persist in the host for extended periods and may reactivate under stress or specific conditions. Uncontrolled breeding and transportation practices frequently contribute to disease outbreaks, resulting in substantial economic losses. For example, in 2007 alone, EI outbreaks in racehorses in Japan and Australia caused economic losses amounting to billions of dollars (11). In recent years, EI has been reported in over 30 countries across Europe, North America, South America, Oceania, Asia, and Africa, indicating its global spread and high incidence rate (12–23). During an EI outbreak, it is crucial to identify infected animals using accurate and rapid diagnostic tests, isolate them promptly for treatment, and vaccinate the remaining healthy population. Thus, there is an urgent need for research into rapid and sensitive detection methods for equine influenza, as well as the development of safe and effective vaccines.

The WOAH-recommended laboratory serodiagnostic methods for equine influenza (EI) include hemagglutination inhibition (HI) and single radial hemolysis (SRH). Among these, the HI assay is a prominent and commonly accepted method used to determine quantitative antibody titers for influenza virus (24). While these methods are efficient and widely used, they are complex to perform and challenging to implement in local quarantine departments. In contrast, the enzyme-linked immunosorbent assay (ELISA) is one of the most widely used serological assays due to its high sensitivity, specificity, convenience, rapidity, and minimal requirements for experimental conditions and personnel, making it highly suitable for the rapid diagnosis of influenza. Although commercial ELISA kits are available globally, there is currently no commercial kit for detecting EIV antibodies in China. This poses significant challenges for equine influenza detection in the country, as kits imported from overseas are prone to damage during long-distance transportation, which can compromise their effectiveness. Additionally, these kits are expensive and have long delivery times. Therefore, the development of a convenient, rapid, and accurate method for detecting EIV antibodies would be highly valuable for clinical diagnosis and epidemiological studies of EI.

The nucleoprotein (NP) is the most abundant protein encoded by segment 5 of the influenza A virus (IAV) genome. It consists of 498 amino acids and has a molecular weight of 56 kDa (25). It would be valuable to mention that anti-NP antibodies can distinguish between influenza types A, B, C, and D, using type-specific ELISAs. NP contains epitopes that are common to all influenza virus subtypes and can elicit cross-reactive antibodies and T-cell responses. As the primary internal viral antigen recognized by CD8+ T cells, NP is widely used in the immunodiagnosis of influenza (26). In this study, we developed a competitive enzyme-linked immunosorbent assay (cELISA) for the detection of EIV NP antibodies. The developed NP-cELISA represents an important serological diagnostic tool with potential applications in the epidemiological investigation of EI.

## MATERIALS AND METHODS

### Plasmids and virus

A total of 16 amino acid sequences of the NP proteins from influenza types A isolated from equine, chicken, duck, human, swine, and canine were aligned using the Megalign software (DNAStar, USA). Then, the EIV NP gene (Genbank EU794544.1, subtype H3N8, XJ lineage) was amplified and cloned into pCAGGS, pET28a, and pVR1012 vector, respectively. The recombinant plasmids pCAGGS-GST-H3N8$_{XJ07}$-NP, pET28a-His-NP, and pVR1012-signalPPT-H3N8$_{XJ07}$NP-6×His were then transformed into DH5α chemically competent cells (Tiangen, China) and purified with FastPure Gel DNA Extraction Mini Kit (Vazyme, China) according to the manufacturer's instructions. NP genes of influenza A from other species were directly synthesized into pCAGGS vector and stored in the laboratory. The EIV (A/equine/XinjiangFuyun/3/07) used in this study was derived from a Chinese epidemic strain (subtype H3N8, XJ lineage) preserved in the laboratory and inactivated with β propiolactone (0.25‰, vol/vol).

### Serum samples

A total of 3,817 sera were used in this study (Table 1). One positive control serum (HVRI-EIV-PS, HI titer 256×) was collected from a horse naturally infected with EIV (subtype H3N8, XJ lineage) in Xinjiang for the NP-cELISA optimization. To establish the Cut off value for the NP-cELISA, 153 serum samples collected from nationwide farms were used, with antibody status confirmed by both the hemagglutination inhibition (HI) test and ID.vet-cELISA (ID.vet, Montpellier, France), including 60 positive and 93 negative samples. Nine positive antisera against other viruses or bacteria were selected for the analytical specificity test. Among them, antisera against equine herpesvirus type 1 (EHV-1), 4 (EHV-4), and equine arteritis virus (EAV) were purchased from National Veterinary Services Laboratories (NVSL, Trent, UK). Positive sera against *Salmonella* Abortusequi (*S.* Abortusequi), *Burkholderia mallei* (*B. mallei*), *Theileria equi* (*T. equi*), *Babesia caballi* (*B. caballi*), avian influenza virus (AIV, H7N4, H9N2), were purchased from National Engineering Center for Veterinary Biological (NECVB, Harbin, China). To further evaluate the detection performance of the NP-cELISA, 119 clinical serum samples and 84 horse serum samples immunized with EIV inactivated antigen (H3N8 subtype, XJ lineage) were collected and tested using this method. Finally, a total of 3,451 equine serum samples collected from 2021 to 2023 across multiple Chinese provinces were analyzed

**TABLE 1** Serum samples employed in this study

| Purpose | Amounts | Name/description | Sample source | Detection techniques |
|---|---|---|---|---|
| NP-cELISA optimization | 1 | Positive control serum against EIV (HVRI-EIV-PS) | Stored in our lab | NP-cELISA /HI |
| Cut off value determination | 153 | Clinical equine sera samples that exhibit negative results in the HI assay and the ID.vet-cELISA (*n* = 93) | Farms in China | NP-cELISA /HI/ ID.vet-cELISA |
| | | Clinical equine sera samples that exhibit positive results in the HI assay and the ID.vet-cELISA (*n* = 60) | | |
| Analytical specificity and sensitivity | 9 | Antisera against *Salmonella* Abortusequi (*S.* Abortusequi), *Burkholderia mallei* (*B. mallei*), *Theileria equi* (*T. equi*), *Babesia caballi* (*B. cabcalli*), avian influenza virus (AIV, H7N4, H9N2) | NECVB[a], China NVSL[b], UK | NP-cELISA |
| | | Antisera against equine herpesvirus 1 (EHV-1), equine herpesvirus 4 (EHV-4), and equine arteritis virus (EAV) | | |
| Evaluation of cELISA | 119 | Clinical sera collected from horses | Farms in China | HI/ ID.vet-cELISA/NP-cELISA |
| | 84 | Sera from experimental horses immunized with EIV | Stored in our lab | NP-cELISA/HI |
| Surveillance on antibody levels of EI | 3,451 | 1,377, 1,260, and 814 sera samples collected in the years of 2021, 2022, and 2023, respectively | Collected from equids in different provinces or regions of China | NP-cELISA |

[a]NECVB, National Engineering Center for Veterinary Biological.
[b]NVSL, National Veterinary Services Laboratories, Inc.

using the NP-cELISA to monitor the seroprevalence of equine influenza (EI) antibodies in China. All serum samples were aseptically collected and transported to the laboratory following standard procedures and stored at −20℃ until further analysis.

## Preparation of the nucleoprotein

To obtain the eukaryotically expressed NP protein for mice immunization, HEK293T cells were plated into 75 $cm^2$ cell culture flasks. Upon reaching 80% confluence, the cells were transfected with 12 μg of the pCAGGS-GST-H3N8$_{XJ07}$-NP plasmid using PolyJet transfection reagent. The cell culture medium was replaced 8 h post-transfection. At 24 h after transfection, the cells were lysed using an alkaline lysis buffer, and the lysate was centrifuged at 12,000 rpm for 10 min to collect the supernatant. The pCAGGS-H3N8$_{XJ07}$NP protein tagged with GST was then purified following the protocol provided with the GST protein purification kit (Beyotime Biotechnology, Shanghai, China) by incubation on a purification column. Subsequently, 5 μL of PreScission Protease was added, and the mixture was incubated at 4℃ overnight. The purified pCAGGS-H3N8$_{XJ07}$-NP protein was finally obtained after washing.

To obtain the prokaryotically expressed pET28a-NP protein for use as a detection antigen, the recombinant plasmid pET28a-*NP* was then transformed into *E. coli* BL21 (Tsingke, Beijing, China) to facilitate overexpression of the protein with an N-terminal His-tag. Briefly, the transformed BL21 cells were induced at 25℃ with an isopropyl-b-D-1-thiogalactopyranoside (IPTG) concentration of 0.5 mM for 8 h. The solubilized His-NP proteins were then purified using HisPur Ni–NTA Resin (Thermo Fisher Scientific, Massachusetts, USA) in accordance with the manufacturer's instructions. The purified proteins with concentrations normalized to 2 mg/mL were conjugated to horseradish peroxidase (HRP) at a molar ratio of 1:1 using a Lightning-Link HRP Conjugation Kit (Innova Bioscience, Cambridge, UK). The labeled products HRP-NP (Horseradish Peroxidase-conjugated Nucleoprotein) were used as the competitive antigen in the cELISA.

## Immunization of mice and monoclonal antibody production

In this study, the plasmid pVR1012-signal-PPT-H3N8$_{XJ07}$***NP***-6×His and eukaryotically expressed NP protein were used as immunogens in the immunization regimen. The immunization protocol involved administering 50 μg of the plasmid to specific-pathogen-free 6-week-old female BALB/c mice (Liaoning Changsheng Biotechnology Co., Ltd., Benxi, China) via intramuscular injection, with a total of four immunizations administered at two-week intervals. Two weeks after the final plasmid immunization, NP protein was administered via intraperitoneal injection at a dose of 100 μg per mouse. Five days after the final booster, the immunized mice were euthanized for hybridoma cell preparation. Splenocytes were taken and fused with myeloma cells SP/20 and then cultured in 96-well plates for 5–7 days. Supernatants from growing hybridoma cells were screened using an indirect ELISA (iELISA) for reactivity to the NP protein. Positive hybridoma clones were diluted according to 1/well and then subcloned in 96-well culture plates until a positive monoclonal cell line was obtained.

Ascites fluid containing mAb was prepared by injecting $5 \times 10^5$ selected hybridoma cells into the abdominal cavity of each mouse. One week prior to harvesting the ascitic fluid, the mice were pre-treated with an intraperitoneal injection of Freund's incomplete adjuvant (Sigma, Ronkonkoma, New York, USA). The ascitic fluid containing the mAb was purified using a protein G perfusion affinity chromatography column (GE Healthcare, Chicago, IL, USA). The purified mAbs were dialyzed in PBS (10 mmol/L) and quantified using a UV spectrophotometer at 280 nm.

## Indirect ELISA

The purified prokaryotically expressed pET28a-NP protein, diluted in phosphate-buffered saline (PBS, PH 7.4), was coated onto 96-well plates (Corning, New York, USA) at a

concentration of 1 µg/mL (100 µL/well) and incubated overnight at 4°C. After incubation, the coated plate was washed twice with PBST (0.05% Tween in 0.01 M PBS) and then blocked with 5% skimmed milk (BD, New Jersey, USA) in PBST (200 µL/well) for 2 h at 37°C. After two additional washes with PBST, 100 µL of hybridoma supernatants was added to each well and incubated for 1 h at 37°C. Sera from NP-immunized mice and non-immunized mice were diluted to 1:200 and used as positive and negative controls, respectively. After subsequent washes, 100 µL of HRP-conjugated goat anti-mouse IgG (Merck, Rahway, USA), diluted 1:5,000 in PBST, was added to the plates and incubated for 30 min at 37°C. Following three times washing, 100 µL of tetramethylbenzidine (TMB) substrate (InnoReagents, Huzhou, China) was added to each well. The plates were incubated at 37°C for 10 min. The reaction was stopped by the addition of 50 µL of 2 M $H_2SO_4$ per well, and the absorbance was measured at 450 nm ($OD_{450nm}$) using a VersaMax Microplate Reader (BioTek, Winooski, USA).

## cELISA

The purified mAb was coated onto the wells of an ELISA plate at a concentration of 100 ng per well and incubated at 4°C for 16 h. Sixty microliters of test sera (diluted 1:8) was mixed with an equal volume of HRP-NP. Subsequently, 100 µL of the mixture was incubated at 37°C for 60 min in the EIV-mAb-coated plates. After incubation, the plates were washed twice with PBST (0.05% Tween-20 in 0.01 M PBS). Following the wash, 100 µL of the TMB was added and the plate was incubated at 37°C for 10 min. The reaction was terminated by adding 50 µL of 2 M sulfuric acid, and the absorbance was measured at 450 nm using a VersaMax Microplate Reader (BioTek, Winooski, USA) (Fig. 1). The serum HVRI-EIV-PS was used as a positive control (PC), and the commercial donor equine serum (DES) (HyClone, Logan, USA) was used as a negative control (NC). Positive and negative controls were included on each ELISA plate. The percent inhibition (PI) of the NP-cELISA was calculated using the formula $(1 - OD_{Sample} /OD_{PC}) \times 100\%$. The Cut off value was determined by testing a total of 153 clinical negative samples using the cELISA and analyzing the resulting PI values.

To compare the capability of our cELISA to detect antibodies against EIV with that of the cELISA (ID.vet, Montpellier, France), we tested 119 clinical sera from China using both assays. All the tests were performed in duplicate, and the obtained data were analyzed in the GraphPad Prism 5 software.

## Hemagglutination inhibition test

The agglutination potency of the viral hemagglutinin (HA) protein is first determined by the hemagglutination assay (HA assay), and then the antigen is diluted to a concentration of 1/4 potency, i.e., 4 HA. In HI test, all sera samples were beforehand treated (16 h) with receptor destroying enzyme (RDE) (Sigma C8772), at a 1:4 dilution, followed by heat inactivation at 56°C for 30 min. Non-specific binding was removed by absorption with erythrocytes according to a WOAH-recommended protocol. Twofold serial dilutions (1:4–1:512) of sera in 25 µL PBS were prepared in V-shaped well plates, and an equal volume of four HA units was added. The mixture was then incubated at room temperature for 30 min. Subsequently, 50 µL of 1% chicken erythrocytes suspended in PBS was added to each well, and the samples were incubated at room temperature for 30 min. The highest serum dilution that exhibits complete hemagglutination inhibition (i.e., red blood cells settle at the bottom of the well as a cell pellet) is considered the HI titer of the serum.

## Immunization of animals with EIV

Four horses confirmed to be negative for equine influenza antibodies, as determined by both ID.vet-cELISA and the HI test, were selected for immunization. The Chinese strain of EIV, A/equine/XinjiangFuyun/3/07 (subtype H3N8, XJ lineage), was inactivated with β-propiolactone (0.25‰, vol/vol) and mixed with an adjuvant (MONTANIDE ISA35) at a 1:1 vol ratio. Each horse was immunized with 4 mL of the inactivated virus-adjuvant mixture.

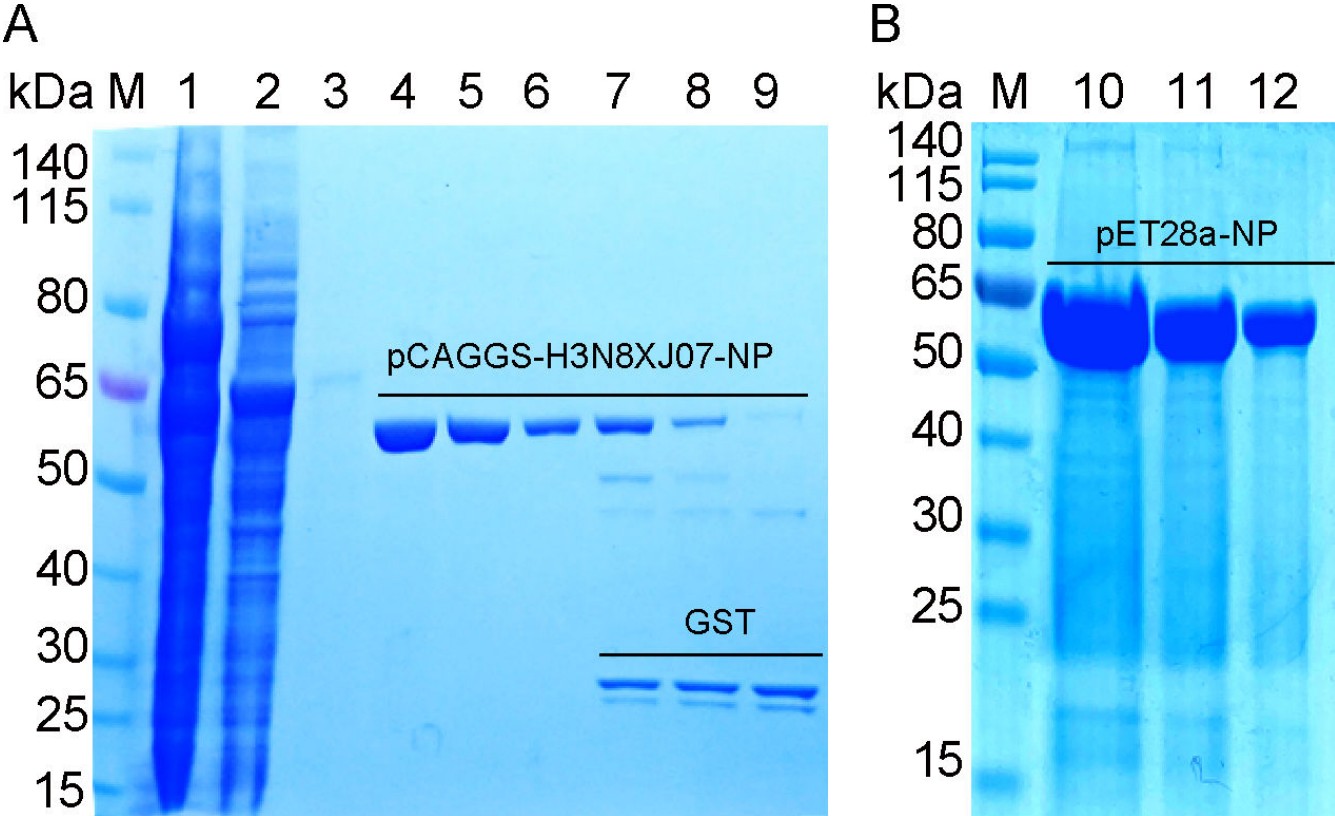

**FIG 1** Expression and purifcation of recombinant pCAGGS-GST-H3N8$_{XJ07}$-NP and pET28a-NP protein. (**A**) SDS–PAGE analysis of pCAGGS-GST-H3N8$_{XJ07}$-NP protein. M, protein marker; lane 1: Supernatant of cell lysate; lane 2: Flow-through; lane 3: Washing; lanes 4–9: Elution. (**B**) SDS–PAGE analysis of purified pET28a-NP protein.

Serum samples was collected from horses before immunization; from day 5 after immunization, with daily collections on days 5–11; every 3 days on days 11–38; and every 2 weeks thereafter. A total of 84 serum (21 × 4) collections were made in this experiment. Seroconversion was then evaluated using the NP-cELISA and HI assays.

## Western blotting

Western blotting was performed as previously described (27) to evaluate the cross-reactivity of the 1E10 monoclonal antibody (mAb) with nucleoprotein (NP) from different influenza A virus strains. Briefly, NP from various influenza A virus species was overexpressed in 293T cells, which were harvested 24 h post-transfection for SDS-PAGE analysis. After electrophoresis, proteins were transferred onto nitrocellulose (NC) membranes. The membranes were blocked with 5% (wt/vol) skim milk in PBS for 2 h at room temperature, followed by incubation with 1E10 monoclonal antibody (25 ng/mL) for 2 h. Subsequently, the membranes were probed with a DyLight800-conjugated anti-mouse IgG secondary antibody (1:5000 dilution, KPL, Hemet, USA) for 1 h at room temperature. Protein bands were visualized using an Odyssey infrared imaging system (LI-COR, UK) at 800 nm.

## Statistical analysis

All experiments were performed at least twice. Data analysis was conducted using GraphPad Prism v.7 (GraphPad Software Inc.). Receiver operating characteristic (ROC) curve performed using 153 clinical sera samples collected from farms in China to determine the cut-off value, sensitivity, and specificity of the cELISA. Cohen's kappa

statistic was used to evaluate the agreement between methods in 2 × 2 contingency tables.

## RESULTS

### Selection of the mAb

The amino acid sequences of the NP proteins from different EIV strains were aligned with homologous sequences from other influenza A virus strains. Amino acid sequence analysis revealed that the NP proteins were highly conserved, showing 91.4%–99.4% identity, and could serve as a universal antigen for detecting all influenza A subtypes (Fig. S1). To develop a rapid diagnostic method for EIV and improve the surveillance and control of this disease, the EIV NP protein was expressed and purified in both HEK293T cells (Fig. 1A) and *E. coli* BL21 (Fig. 1B). To reduce the screening of hybridoma cell lines secreting the GST-tag, the pET28a-His-NP protein with His-tag was used to screen a panel of mAbs specific to this antigen. These mAbs were produced using hybridoma technology. After screening, four clones demonstrating the strongest binding activity to EIV NP were obtained and designated as 1E5, 1E10, 5B8, and 6A12.

To identify the mAb with the highest competitive activity for cELISA, four mAbs were coated onto ELISA plates at varying dilutions, followed by simultaneous incubation of the test serum samples and HRP-NP antigen in a competitive binding reaction (Fig. 2). Among them, 1E10 exhibited the strongest inhibitory effect, with $OD_{Negative}/OD_{Positive}$ (N/P) values of 45.08, 33.92, 26.35, and 14.19 at the four tested dilutions, demonstrating its ability to effectively differentiate between positive and negative samples (Fig. 3). Finally, the good cross-specificity of 1E10 with different species of influenza A virus NP proteins was further confirmed by western blotting (Fig. 4). Based on these findings, 1E10 was selected to develop the cELISA for EIV detection, where it was utilized to capture HRP-NP.

### Establishment of the NP-cELISA

A NP-cELISA was performed using purified 1E10 MAb adsorbed to the plate. Then, to optimal detection efficacy, the most suitable commercial serum diluent (Solution I, II, III, [PR-SS-001 to 003], InnoReagents, Hubei, China) and the optimal serum dilution were determined. A twofold serial dilution (from 1 to 64) of the EIV positive serum was performed using three different commercial serum diluents. Three different commercial serum diluents were used to perform a twofold serial dilution (ranging from 1 to 64) of the equine influenza virus (EIV) positive serum. Subsequently, the absorbance values at 450 nm ($OD_{450}$) of the diluted serum samples were measured and recorded. Then, the ratio of the absorbance of the N/P value was calculated to evaluate the performance of each dilution condition. The experimental results showed that the highest N/P value was obtained when Solution I was used as the diluent and the serum dilution was 8.

The 153 sera verified by HI and ID.vet-cELISA (93 naive sera and 60 infected sera) were further evaluated using the newly developed NP-cELISA to determine the Cut off value, sensitivity, and specificity of the assay. The percent inhibition values for each sample were then calculated and visualized in an interactive dot plot (Fig. 5A). Additionally, a receiver operating characteristic (ROC) analysis was performed to determine the Cut off value balancing assay sensitivity and specificity (Fig. 5B). From the ROC analysis, the area under the curve (AUC) was determined to be 1.000 (95% confidence interval: 1.000 to 1.000). Furthermore, when the Cut off value was set at 55% for the NP-cELISA, the test sensitivity and specificity were both as 100.00% (95% CI, 0.9398–1.0000) and 100.00% (95% CI, 0.9603–0.9949), respectively. Based on these results, the cut-off value of percent inhibition (PI) was set at 55%. Consequently, serum samples with a PI of <55% were classified as negative, whereas those with a PI of ≥55% were classified as positive.

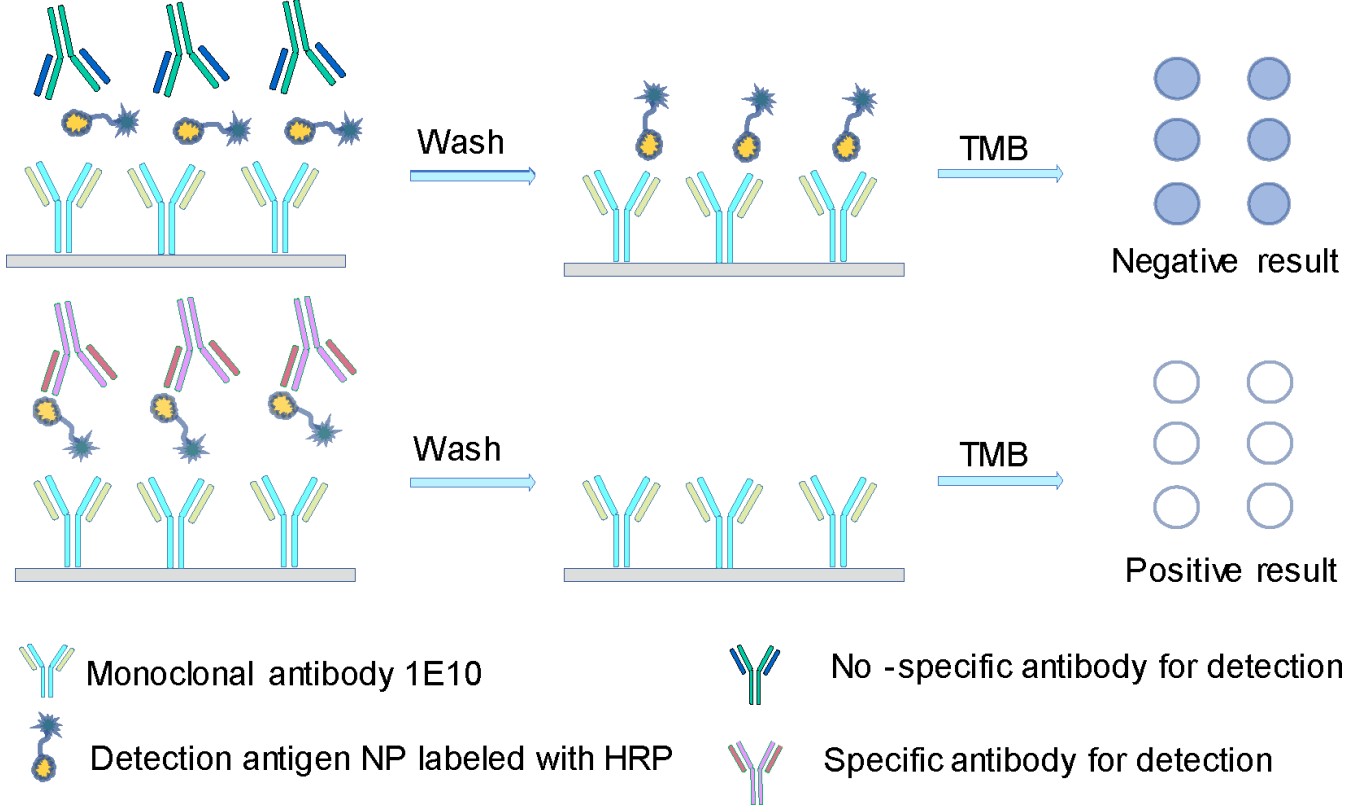

**FIG 2** The detecting principle of the cELISA. Anti-EIV serum reacts specifically with NP-HRP in the liquid phase, and competitively blocks the binding of 1E10 mAb precoated in the plate with its recognized epitope on NP-HRP in the solid phase. The conjugation is washed away during the cELISA procedure. Little or no color is visible in the positive samples. Conversely, negative samples have a darker color.

## Evaluation of the specificity, sensitivity, and reproducibility of the NP-cELISA

The specificity of the NP-cELISA for detecting antibodies against the NP was rigorously evaluated using positive sera known to contain antibodies against EHV-1, EHV-4, EAV, *S.* Abortusequi, *B. pseudomallei*, *T. equi*, and *B. equi*. The NP-cELISA produced positive signals exclusively for samples containing EIV (H3N8), AIV (H7N4, H9N2) (Fig. 5C). Furthermore, no cross-reactivity was detected with antibodies specific to the aforementioned pathogens, confirming the high cross-specificity of the assay for Influenza A viruses NP antibodies.

To evaluate the sensitivity of the method, the EIV positive control serum (HI titer 256×) was subjected to a twofold serial dilution, ranging from 1× to 2,048×, and the maximum dilution that still resulted in a positive value in the NP-cELISA was a dilution of 256× (Fig. 5D). To further validate inter-batch reproducibility, the positive serum sample was tested with three replicates in three independent batches of HRP-NP conjugate, demonstrating comparable sensitivity across all batches (Fig. 5D).

## Performance of the NP-cELISA compared with ID.vet-cELISA in field samples

A total of 119 serum samples were tested using both assays. Compared to ID.vet-cELISA, the newly developed NP-cELISA demonstrated a sensitivity of 100% (66/66) and a specificity of 67.9% (36/(36 + 17)). The overall agreement between the two assays was 85.7% ((66 + 36)/119) (Table 2). To investigate the discordance between the serological results of the NP cELISA and ID.vet-cELISA, the WOAH-recommended HI assay was used for further confirmation. The coincidence rates of NP-cELISA and ID.vet-cELISA with HI were 87.4% (77 + 27)/119) and 78.2% (63 + 30)/119), respectively (Table 3). Among the 17 samples that were positive in the NP-cELISA but negative in the ID.vet-cELISA, 14 tested

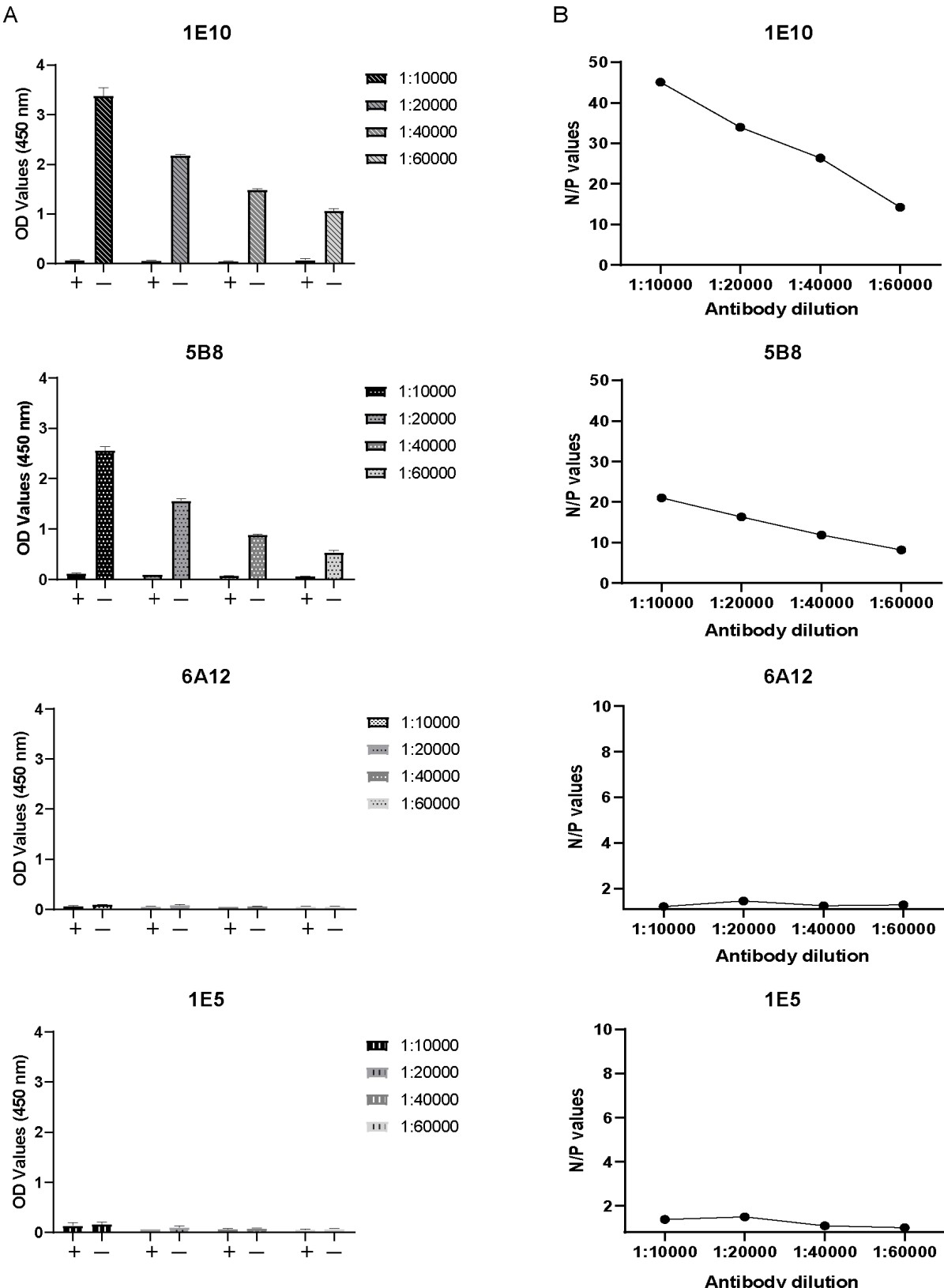

**FIG 3** Screening coating mAbs for the development of cELISA. Conducted an experiment where antibodies were diluted at four different concentrations. (**A**) The OD450 values of positive and negative serum detected by cELISA coated with different monoclonal antibodies. (**B**) The N/P values of positive and negative serum detected by cELISA coated with different monoclonal antibodies.

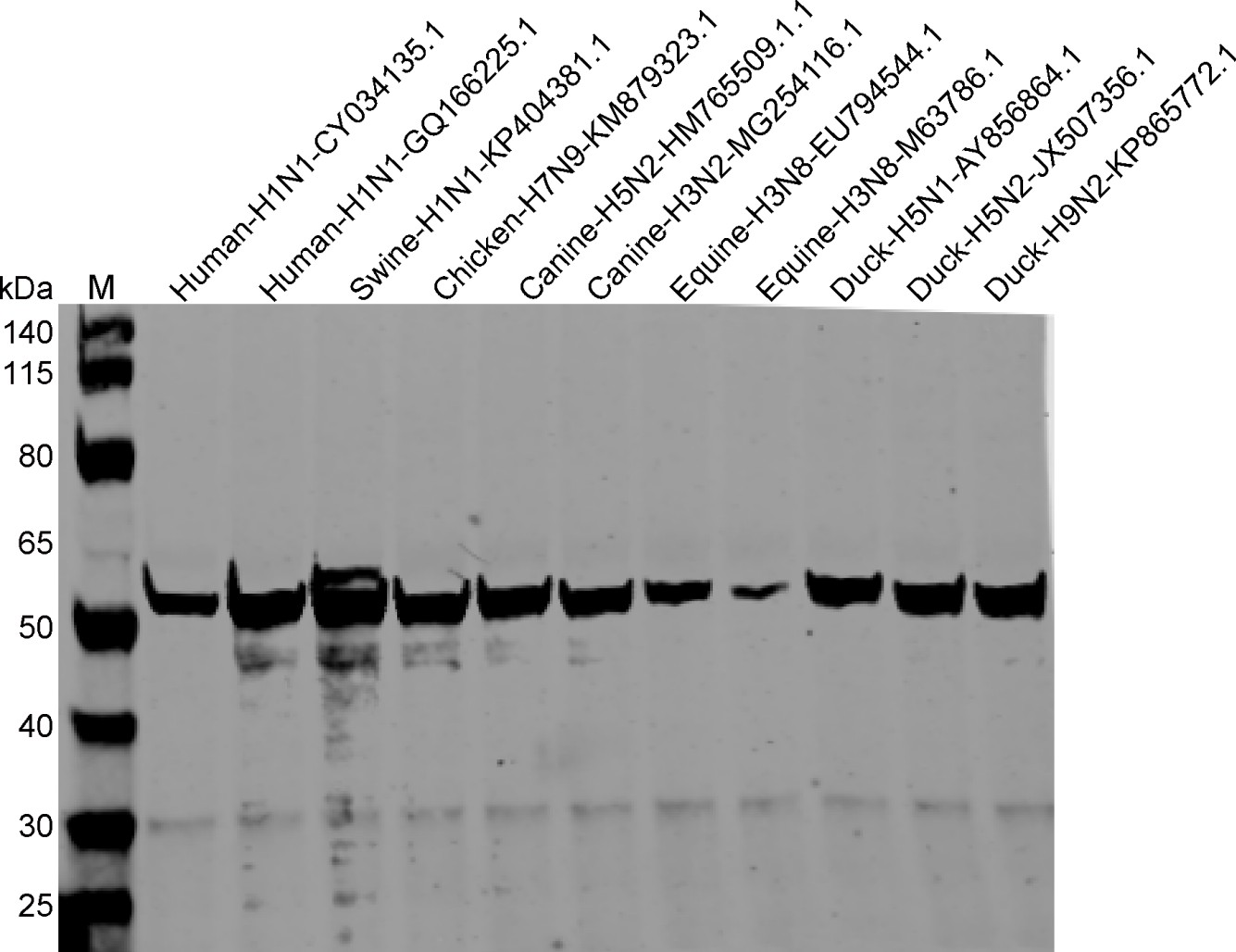

**FIG 4** Verification of the cross-reactivity of 1E10 monoclonal antibody with NP proteins of influenza A viruses of different species by western blotting.

positive and 3 tested negative in the HI assay. The concordance between the NP-cELISA and HI assay results was higher than that between ID.vet-cELISA and HI assay results.

## Performance of the NP-cELISA compared with HI in immunized horse samples

To further evaluate the sensitivity of the developed NP-cELISA, we analyzed the serum antibody dynamics in experimentally immunized animals. Four horses (#1, #2, #3, and #4) were immunized simultaneously with inactivated equine influenza virus, and blood samples were collected daily. Using the NP cELISA, seroconversion was detectable on day 5 for samples #1(Fig. 6A), #2(Fig. 6B), and #3(Fig. 6C), while sample #4 showed seroconversion on day 11 (Fig. 6D). Throughout the monitoring period, starting from the first day post-immunization, serum antibody titers increased and remained elevated without decline up to day 94. In contrast, using the HI assay, seroconversion for samples #1, #2, and #3 (Fig. 6A–C) was observed on day 7, while sample #4 (Fig. 6D) turned positive on day 8. During the monitoring period, serum titers began to rise from the 6th day post-immunization, peaked on the 10th day, and subsequently declined, with sample #4 becoming seronegative by day 66.

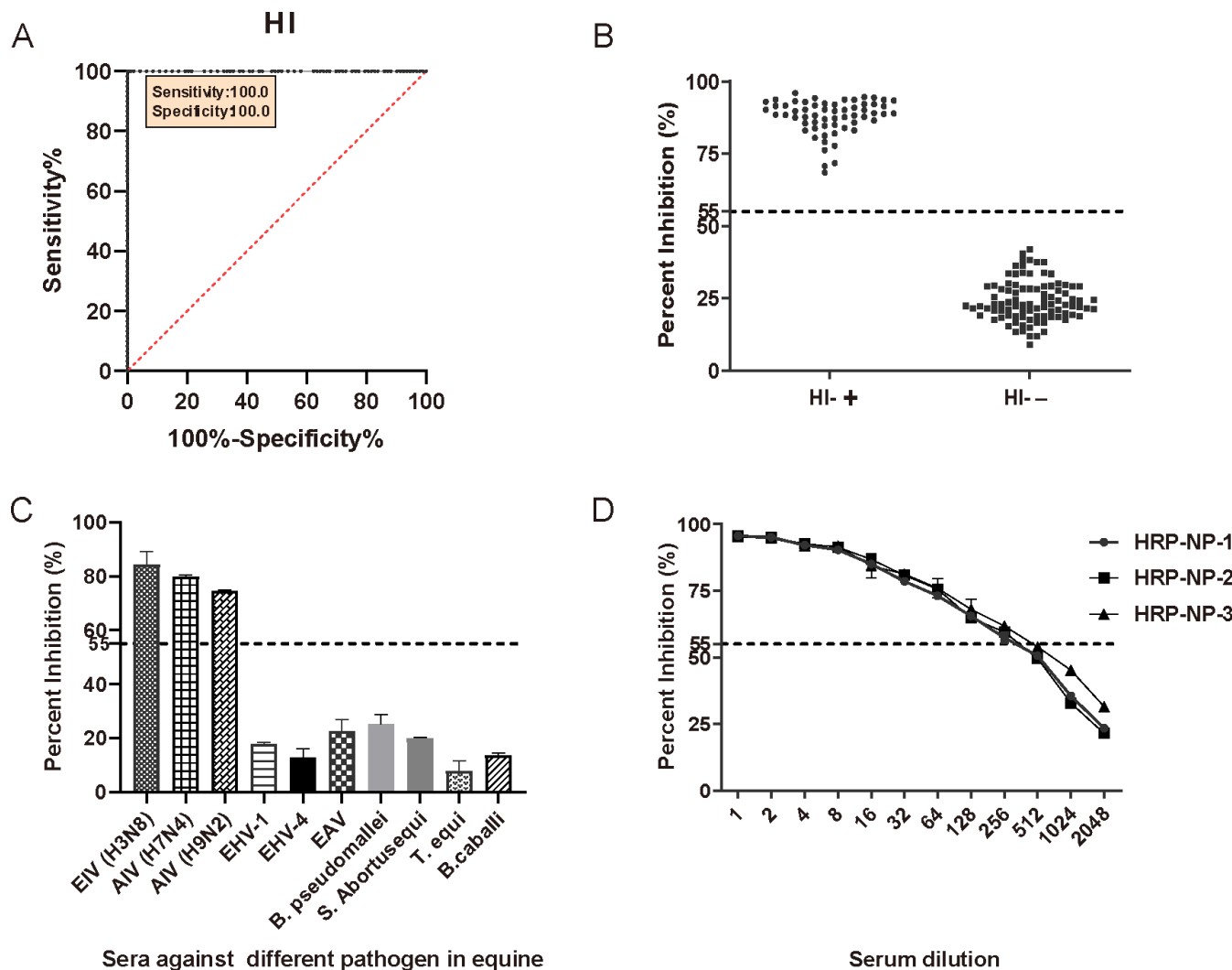

**FIG 5** Evaluation of the NP-cELISA. (**A and B**) ROC analysis for the NP-cELISA using EIV-negative sera (*n* = 93) and EIV-positive sera (*n* = 60). (**C**) Specificity and (**D**) sensitivity of the NP-cELISA. The horizontal dotted line represents the cut-off value.

## Epidemiological trends in equine influenza in China from 2021 to 2023

The developed NP-cELISA was used to analyze 3,451 equine serum samples collected from 21 provinces and regions across China between 2021 and 2023 (Fig. 7). Among these, 1,310 samples tested positive, resulting in a national seroprevalence rate of 37.69% (Table 4). Specifically, the seropositivity rates for EI were 36.75% (506/1,377) in 2021, 40.08% (505/1,260) in 2022, and 36.73% (299/814) in 2023.

China encompasses a vast geographical area, which can be divided into seven distinct regions—the Northeast, North, Northwest, East, Central, Southwest, and South—based on geographical location, natural features, and climatic patterns. The seroprevalence rates varied significantly across these regions, ranging from 20.74% (112/540) in the Northeast to 51.61% in the North.

**TABLE 2** Comparison of the results for NP-cELISA with ID.vet-cELISA for testing 119 serum samples

| | | ID.vet-cELISA | | Kappa | 95% CI | *P* value |
|---|---|---|---|---|---|---|
| | | + | − | | | |
| NP-cELISA | + | 66 | 17 | 0.65 | 0.53–0.77 | < 0.001 |
| | − | 0 | 36 | | | |

**TABLE 3** Comparison of the results for cELISA and ID.vet-cELISA with HI for testing 119 serum samples

| | | HI | | Kappa | 95% CI | *P* value |
|---|---|---|---|---|---|---|
| | | + | − | | | |
| NP-cELISA | + | 77 | 6 | 0.687 | 0.573–0.801 | <0.001 |
| | − | 9 | 27 | | | |
| ID.vet-cELISA | + | 63 | 3 | 0.505 | 0.366–0.644 | <0.001 |
| | − | 23 | 30 | | | |

## DISCUSSION

Influenza viruses are negative-sense, segmented RNA viruses characterized by antigenic drift and shift (4, 28). The antigenic variability of influenza A viruses (IAVs) poses significant challenges for diagnosis and control, underscoring the need for accurate and rapid diagnostic methods. A key challenge in developing ELISA-based diagnostics for IAVs is selecting targets that enable broad-spectrum detection and high sensitivity. In vaccine and diagnostic design, viral surface proteins are often targeted due to their critical role in virus-host cell interactions (29). Among these, hemagglutinin (HA) and neuraminidase (NA) are the primary surface proteins, essential for virus adsorption, host cell invasion, and viral particle release. HA, owing to its high surface exposure and strong immunogenicity, is widely used in vaccines and diagnostic testing (30). However,

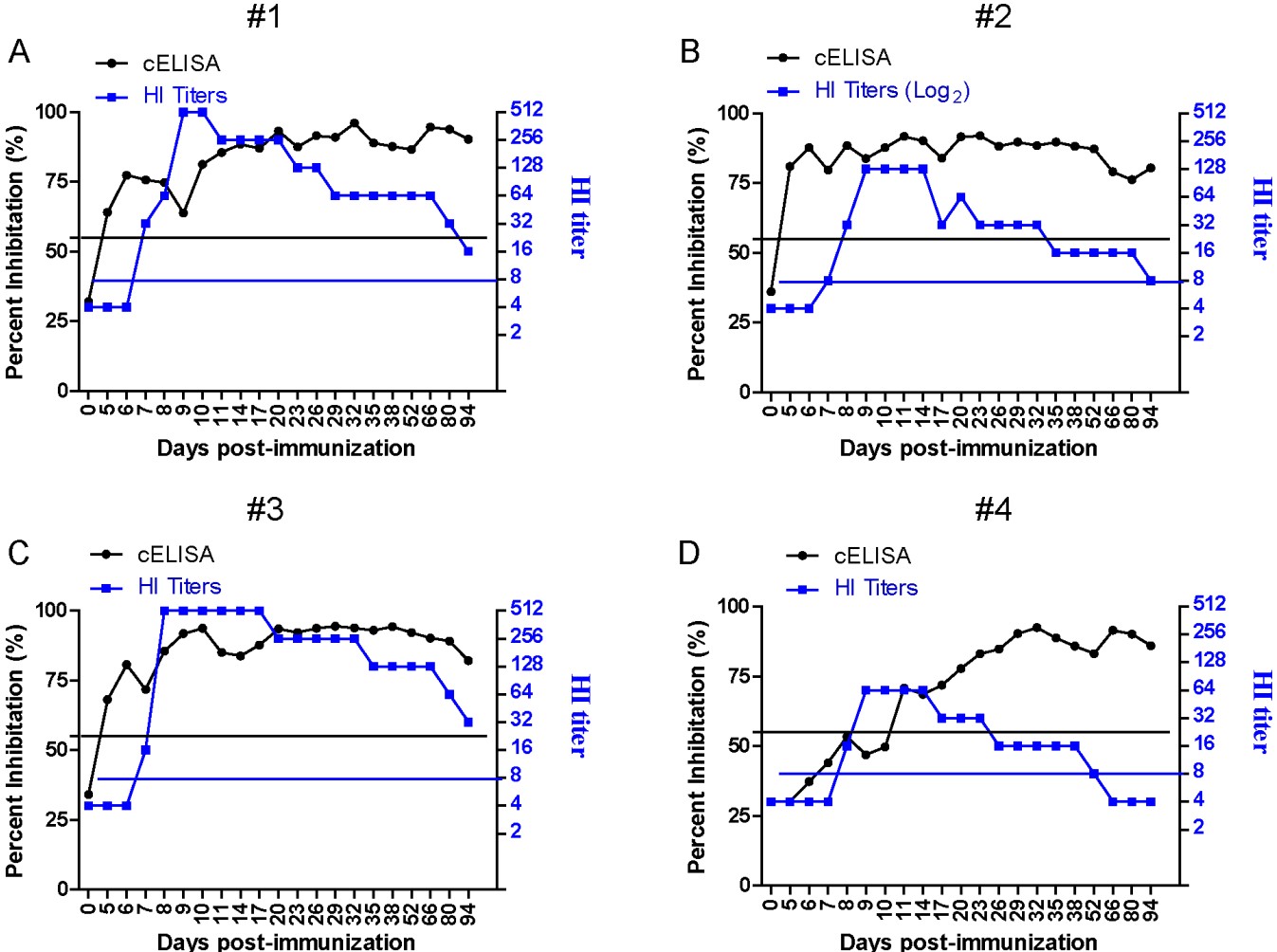

**FIG 6** The detection of the antibody dynamics to EIV was carried out using the NP cELISA and HI test with the sera from experimentally immunized horses (A–D). The black horizontal dotted line represents the cut-off value of the cELISA. The blue horizontal dotted line represents the cut-off value of the HI test.

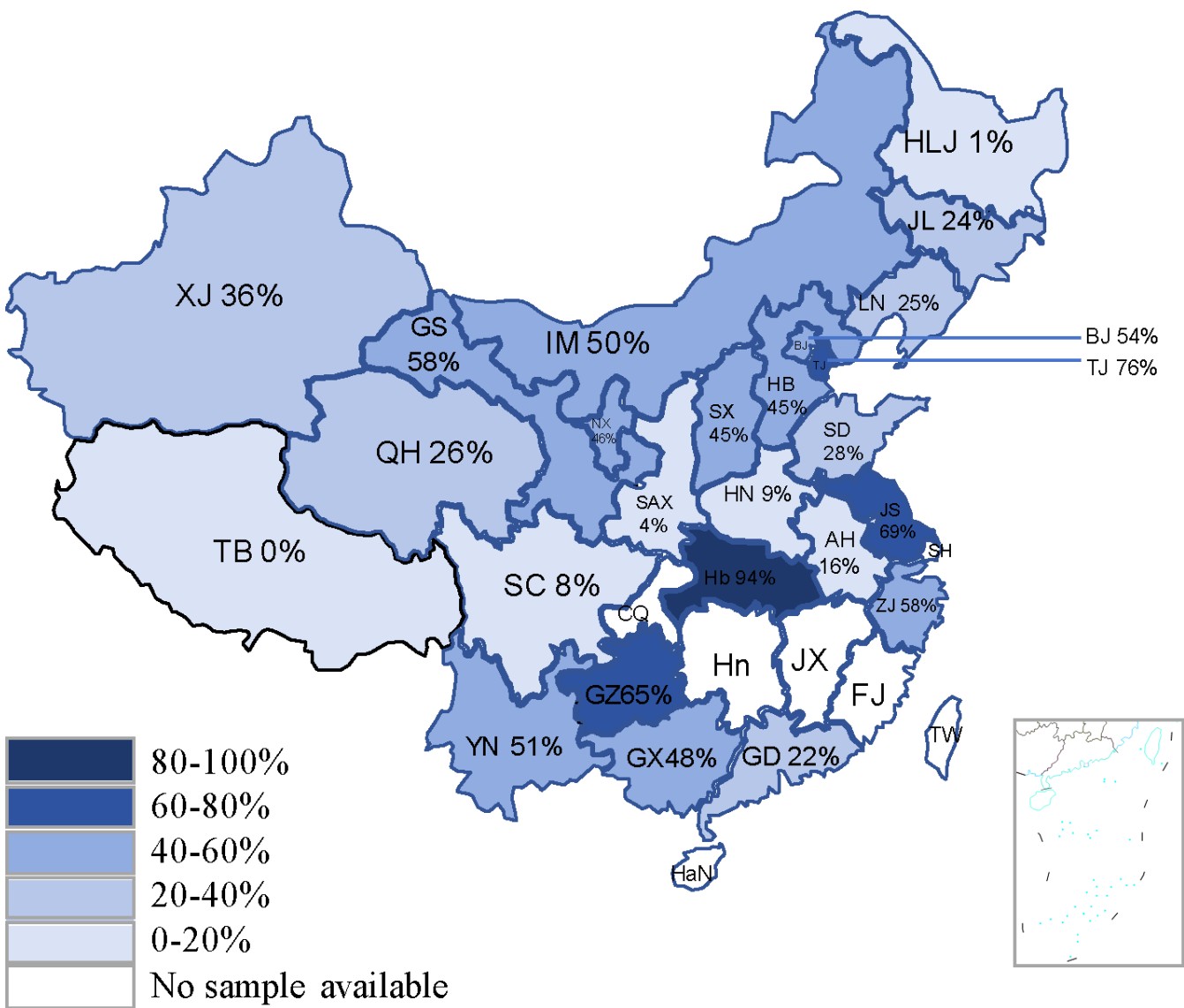

**FIG 7** Application of the cELISA to detect anti-EIV antibodies in China between 2021 and 2023.The intensity of color represents the level of seropositivity for equine influenza antibodies, with darker shades indicating higher seropositivity rates and lighter shades indicating lower rates.

nucleoprotein (NP) is increasingly considered an alternative target for antibody detection due to its high conservation and ability to distinguish influenza A from other types (25, 31). Unlike HA, NP is less affected by antigenic drift and shift, making it a more reliable target for broad-spectrum diagnostics (31, 32). In summary, while HA is often targeted for its role in viral infection, NP's conservation and stability make it a valuable alternative for influenza virus detection, particularly for broad coverage of diverse strains. This approach improves the universality and reliability of diagnostic tests.

The main advantage of cELISAs over iELISAs is that the cELISA assays for the detection of antibodies enable the testing of samples from different animal species (33). Therefore, the major goal of this study was to develop a rapid and simple mAb-based cELISA to detect antibodies against EIV in different animal species. Monoclonal antibodies (mAbs) with competitive activity are crucial for developing competitive ELISA (cELISA). In this study, a mAb preparation process was employed, involving initial immunization with a recombinant plasmid containing the NP gene, followed by booster immunization using eukaryotically expressed NP protein. The NP was used solely for booster immunization prior to fusion. The eukaryotic expression system produces proteins closer to the

**TABLE 4** Serological surveillance of equine influenza virus in China from 2021 to 2023[a]

| Areas | Provinces/regions | Positive | | | Average | |
|---|---|---|---|---|---|---|
| | | 2021 | 2022 | 2023 | | |
| Northeast China | Heilongjiang (HLJ) | 1.11% (1/90) | N/A | N/A | 1.11% (1/90) | 20.74% (112/540) |
| | Jilin (JL) | 0% (0/90) | 48.89% (44/90) | N/A | 24.44% (44/180) | |
| | Liaoning (LN) | 32.22% (29/90) | 31.11% (28/90) | 11.11% (10/90) | 24.81% (67/270) | |
| North China | Beijing (BJ) | 76.67% (69/90) | 42.22% (38/90) | 44.44% (40/90) | 54.44% (147/270) | 51.61 (449/870) |
| | Tianjin (TJ) | N/A | 75.56% (68/90) | N/A | 75.56% (68/90) | |
| | Shanxi (SX) | 28.89% (26/90) | 37.78% (34/90) | 80.00% (48/60) | 45.00% (108/240) | |
| | Inner Mongolia (IM) | N/A | N/A | 50.00% (45/90) | 50.00% (45/90) | |
| | Hebei (HB) | 42.22% (38/90) | 47.78% (43/90) | N/A | 45.00% (81/180) | |
| Northwest China | Shaanxi (SAX) | 4.44% (4/90) | 3.33% (3/90) | N/A | 3.89% (7/180) | 34.58% |
| | Gansu (GS) | 66.676% (60/90) | 50.00% (45/90) | N/A | 58.33% (105/180) | (249/720) |
| | Qinghai (QH) | N/A | N/A | 25.56% (23/90) | 25.56% (23/90) | |
| | Ningxia (NX) | 51.11% (46/90) | 40.00% (36/90) | N/A | 45.56% (82/180) | |
| | Xinjiang (XJ) | N/A | 35.56% (32/90) | N/A | 35.56% (32/90) | |
| East China | Jiangsu (JS) | N/A | N/A | 68.89% (62/90) | 68.89% (62/90) | 42.50% |
| | Zhejiang (ZJ) | 57.78% (52/90) | N/A | N/A | 57.78% (52/90) | (153/360) |
| | Anhui (AH) | N/A | N/A | 15.56% (14/90) | 15.56% (14/90) | |
| | Shandong(SD) | N/A | N/A | 27.78% (25/90) | 27.78% (25/90) | |
| Central China | Henan (HN) | 0% (0/90) | 21.67% (13/60) | N/A | 8.66% (13/150) | 51.5% |
| | Hubei Hb) | N/A | N/A | 94.11% (32/34) | 94.11% (32/34) | (45/184) |
| Southwest China | Yunnan (YN) | 74.44% (67/90) | 26.67% (24/90) | N/A | 50.56% (91/180) | 41.09% |
| | Guizhou (GZ) | 69.23% (45/65) | 62.22% (56/90) | N/A | 65.16% (101/155) | (196/477) |
| | Sichuan (SC) | 7.69% (4/52) | N/A | N/A | 7.69% (4/52) | |
| | Tibet (TB) | N/A | N/A | 0% (0/90) | 0% (0/90) | |
| South China | Guangdong (GD) | 22.22% (20/90) | N/A | N/A | 22.22% (20/90) | 39.26% |
| | Guangxi (GX) | 50.00% (45/90) | 45.56% (41/90) | N/A | 47.78% (86/180) | (106/270) |
| Average | | 36.75% (506/1377) | 40.08% (505/1,260) | 36.73% (299/814) | 37.96% (1,310/3,451) | |

[a]Not available.

true conformation of NP proteins than the prokaryotic system. Plasmid immunization combined with protein immunization strategy reduce the use of NP proteins (Used only once during immunization) compared to conventional methods. DNA primary immunization promotes B-cell affinity maturation by prolonging the germinal center response through sustained antigen expression, while eukaryotic protein enhancement selectively expands high-affinity B-cell clones through high concentrations of natural antigens, both synergistically driving efficient antibody affinity maturation (34). Among the four mAbs screened, 1E10 exhibited the strongest binding capacity to NP, outperforming the other three mAbs (1E5, 5B8, and 6A12). Additionally, 1E10 showed superior ability to distinguish between positive and negative samples (Fig. 3). Based on these results, 1E10 was selected as the most suitable mAb for developing the NP-cELISA.

The NP-cELISA was developed using the purified mAb coating the plate, followed by simultaneous incubation with a mixture of HRP-conjugated NP antigen and serum samples. Unlike conventional solid-phase ELISAs that use labeled mAbs, this method relies on liquid-phase interactions between the serum sample and HRP-pET28a-NP, which is more favorable for antigen-antibody reactions. This approach may provide a more efficient ELISA development strategy at reaction times (one-step reaction). The developed NP-cELISA procedure is a one-step reaction, completing 96 tests in just 70 min. Compared to a previous Sandwich ELISA (35), this method is 80 min faster and outperforms ID.vet-cELISA by 30 min, saving significant time and labor, making it ideal for high-throughput screening.

To verify whether the NP-cELISA developed can detect antibodies raised after infection by diverse influenza subtypes in different species. First, multiple sequence alignment analysis showed that the amino acid sequences of different influenza A virus

NP proteins were highly conserved (Fig. S1). We further verified the cross-reactivity of 1E10 with other influenza A viruses and found that this monoclonal antibody could react with all the influenza A viruses NP selected in this study (Fig. 4). The cross-specificity of the developed NP-cELISA was further validated using a series serum against EIV (H3N8) and AIV (H7N4, H9N2). 1E10-based NP-cELISA might be used for the detection of antibodies against different subtypes of influenza A viruses. The broader applicability of the NP-cELISA for influenza type A detection across multiple subtypes was needed further validated in future study due to the unavailability of clinical samples of other species.

The sensitivity of the NP-cELISA was validated using a large number of field serum samples from different species and immunized horse sera. The stability of the HRP-NP protein and its labeling efficiency will affect both the repeatability and sensitivity of the assay results. Inter-batch reproducibility was validated by testing the positive serum in triplicate across three independent HRP-NP conjugate batches, showing consistent sensitivity (256×, equal to HI titer). Among 119 clinical samples, the positive detection rates were comparable between NP-cELISA (69.7%, 83/119) and HI assay (72.3%, 86/119), both significantly higher than ID.vet-cELISA (55.5%, 66/119). Using the WOAH-recommended HI assay as the reference standard, the analysis revealed superior concordance between NP-cELISA and HI (87.4%) compared to that between ID.vet-cELISA and HI (78.2%) (Table 3).

To further assess sensitivity of the NP-cELISA, four horses were immunized, and antibody titers were monitored using both HI and the NP-cELISA. The NP-cELISA detected seroconversion earlier than HI in 3 out of 4 horses (Fig. 6), likely due to differences in the targeted antigens: NP for the NP-cELISA and HA for HI. Variations in antibody production timing and levels, as well as individual immune responses, may also contribute to this discrepancy (36, 37). In addition, the difference in analytical sensitivity between the two tests can also affect the results. In summary, the developed NP-cELISA demonstrates promising performance in monitoring antibody kinetics in immunized horses, highlighting its potential for broad application. As horses are a key model for studying EIV pathogenesis and vaccine development, this cELISA provides a valuable tool for monitoring EIV antibody responses and vaccine efficacy.

To investigate the prevalence of equine influenza (EI) in China, we conducted a serological survey using the NP-cELISA to detect EIV antibodies in equine serum samples. A total of 3,451 samples were collected from 21 provinces and regions across China between 2021 and 2023. The seroprevalence rates were consistently high, recorded at 36.75%, 40.08%, and 36.73% for each respective year (Table 4), indicating widespread EIV infection in China's equine population. When analyzed by region, the Northeast showed the lowest seroprevalence (20.74%), likely due to its arid and cold climate, which may hinder EIV transmission. Seasonal influences on EIV dissemination were suspected, but the lack of detailed sampling information limited this analysis. Notably, Guizhou, Hubei, and Jiangsu exhibited disproportionately high seroprevalence, potentially linked to local equestrian events and horse trading activities. In summary, equine movement appears to be a key factor in EI prevalence, highlighting the need for targeted surveillance in high-mobility regions. In China, EI is not subject to mandatory vaccination, and few legally approved EI vaccines are available. Therefore, the seroprevalence rates reflect natural infection conditions. This study represents the first large-scale serological survey of EIV in China, providing critical insights into EI epidemiology.

In summary, this study successfully generated NP mAbs against EIV and developed a NP-cELISA using the mAb 1E10 for detecting EIV-specific antibodies. The NP-cELISA demonstrated high specificity and sensitivity and showed strong agreement with established methods like ID.vet and the HI test, confirming its reliability as a diagnostic tool. These results suggest that the NP-cELISA can serve as an alternative to ID.vet-cELISA for EIV antibody detection. This NP-cELISA provides a simple, cost-effective, and valuable tool for serodiagnosis and EI surveillance. Its development could significantly contribute to EI prevention and control by enabling efficient monitoring of antibody prevalence

in equine populations, supporting timely control measures and informed vaccination strategies.

## ACKNOWLEDGMENTS

This study was funded by grants from the Nature Science Foundation of Heilongjiang Province, China (TD2022C006), the National Key Research and Development Program of China (No. 2021YFD1800500), Xinjiang Talent Development Fund (ZZYD2023010), and Tianchi Talent Introduction Plan (IWA2023).

Y.Y.: Writing—Original draft, Investigation, Data curation. K.G.: Software, Writing—review & editing. Z.Z. and L.X.: Methodology, Investigation. W.G., M.D., W.L., and S.L.: Resources, Formal analysis. Z.Z., X.C., and Y.W.: Resources, Visualization, Conceptualization. Z.H.: Writing—original draft, Validation. X.W.: Writing—review & editing, project administration, funding acquisition.

## AUTHOR AFFILIATIONS

[1]State Key Laboratory for Animal Disease Control and Prevention, Harbin Veterinary Research Institute, the Chinese Academy of Agricultural Sciences, Harbin, Heilongjiang, China

[2]Harbin Guosheng Biotechnology Co. Ltd., Harbin, China

[3]University of Wisconsin-Madison School of Medicine and Public Health, Madison, Wisconsin, USA

[4]Institute of Western Agriculture, the Chinese Academy of Agricultural Sciences, Changji, Xinjiang, China

## AUTHOR ORCIDs

Yan Yang  http://orcid.org/0000-0002-3862-2003
Kui Guo  http://orcid.org/0000-0001-6274-1857
Zhenyu Zhang  http://orcid.org/0000-0003-4974-1400
Zhe Hu  http://orcid.org/0000-0003-2293-0105
Xiaojun Wang  http://orcid.org/0000-0003-4521-4099

## FUNDING

| Funder | Grant(s) | Author(s) |
|---|---|---|
| Tianchi Talent Introduction Plan | IWA2023 | Xiaojun Wang |
| The National Key Research and Development Program of China | No. 2021YFD1800500 | Zhe Hu |
| Xinjiang Talent Development Fund | ZZYD2023010 | Xiaojun Wang |
| The Nature Science Foundation of Heilongjiang Province | TD2022C006 | Xiaojun Wang |

## ETHICS APPROVAL

All animal experiments in this study received approval from the Animal Ethics Committee of Harbin Veterinary Research Institute of the Chinese Academy of Agricultural Sciences, under approval number 231023-04-SW.

## ADDITIONAL FILES

The following material is available online.

Supplemental Material

**Fig. S1 (Spectrum00939-25-s0001.pdf).** Amino acids sequence alignment of NP from influenza types A (Equine, Chicken, Duck, Human Swine, Canine).

## Open Peer Review

**PEER REVIEW HISTORY (review-history.pdf).** An accounting of the reviewer comments and feedback.

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
