## [Reviewer comments · Microbiology Spectrum]

Microbiology Spectrum

Development and application of a NP-cELISA for the detection of nucleoprotein antibodies of equine influenza virus

Yan Yang, Kui Guo, Ling Xu, Wei Guo, Mingqi Dong, Wen Liu, Shuaijie Li, Zenan Zhang, Xiaoyu Chu, Yaoxin Wang, Zhenyu Zhang, Zhe Hu, and Xiaojun Wang

Corresponding Author(s): Xiaojun Wang, State Key Laboratory for Animal Disease Control and Prevention, Harbin Veterinary Research Institute, the Chinese Academy of Agricultural Sciences, Harbin, China

Review Timeline:

Submission Date:	April 2, 2025
Editorial Decision:	April 24, 2025
Revision Received:	June 19, 2025
Editorial Decision:	July 22, 2025
Revision Received:	July 23, 2025
Accepted:	July 25, 2025

Editor: Yunyu Chen

Reviewer(s): Disclosure of reviewer identity is with reference to reviewer comments included in decision letter(s). The following individuals involved in review of your submission have agreed to reveal their identity: Carmen Guzmán-Bracho (Reviewer #1); Mohammed FILALI (Reviewer #3)

Transaction Report:

DOI: <https://doi.org/10.1128/spectrum.00939-25>

Re: Spectrum00939-25 (Development and application of a HRP-pET28a-NP and mAb-based cELISA for the detection of nucleoprotein antibodies of equine influenza virus)

Dear Prof. Xiaojun Wang:

Thank you for the privilege of reviewing your work. Below you will find my comments, instructions from the Spectrum editorial office, and the reviewer comments.

Your manuscript has now been reviewed by expert reviewers for the journal. The reviewers have raised some important issues and have made many useful suggestions for its improvement. Based on these comments (attached below), as well as my own review, we would be happy to receive a revised version of the manuscript that is responsive to the reviewer comments. Nucleoprotein is usually used as a universal antigen to detect type-specific (Type A, B, C or D) influenza virus. I feel the study would immensely benefit if the authors can show that the cELISA developed can detect antibodies raised after infection by diverse influenza subtypes in different species. The study otherwise is of less significance. In addition, the Materials and Methods section should be carefully revised.

Revision Guidelines

Sincerely,
Yunyu Chen
Editor
Microbiology Spectrum

Reviewer #1 (Comments for the Author):

The manuscript Development and application of a HRP-pET28a-NP and mAb-based cELISA for the detection of nucleoprotein antibodies of equine influenza virus was reviewed. The study included different experimental stages to develop, evaluate performance and apply a new serological test in population diagnosis.

After reading this manuscript I had identify interesting results. So, to contribute to the authors' improvement of this manuscript, I make the following comments.

Lines 33 and 34. OUR ESSAY? Which one? ... A commercial kit, which one?

It is considered convenient to inform readers of the performance parameters of the Hemagglutination Inhibition test as it is the reference test with which the other tests are compared. In other words, the HI's validity as reference test will be strengthened by revealing its sensitivity and specificity.

Line 34. In this manuscript, it was omitted mention of the performance of the reference test inhibition of hemagglutination, with which the tests under evaluation are compared.

Line 101. This section should be rewritten by adding some paragraphs and rearranging existing paragraphs to give greater clarity to the manuscript.

The study design is multi-stage, as is shown in Table 1, indicated up to line 110. The flowchart created to realize the development and evaluation of serological tests is strongly recommended to be included at the beginning of this section of Material and Methods by a clear and brief text. Likewise, it is recommended that after describing the methodological design, the techniques used must be included in a structure that allows readers to identify which stage they correspond to and, in a short sentence, justify their use, that is, indicate what the authors intended to obtain when applying each technique. It is also highly recommended to make explicit in the section Serum samples, the characteristics of the samples that were used in each stage, since their mention in this manuscript is confusing and inconsistent for the reader: line 33=119, line 110=153, line 120 = 84, line 122=3,451.

Line 107. Five groups of samples are identified. You could indicate this at the beginning of this section and perhaps number them to make it clearer to the reader. It is also important to clarify whether ALL samples were processed using the three techniques, iELISA, cELISA, and haemagglutination inhibition.

Line 108. Considering the value of this serum in the present study, it will be convenient to mention the characteristics of this serum (origin, tests used for classification as positive, storage, etc.).

Line 113. The ID.vet test corresponds to an ELISA test (line 204). Since the first mention in the abstract and this line 113, I recommend indicating that it is an ELISA test.

Lines 125, 149, 169, 128. I suggested writing the full name and abbreviations in parentheses in the subtitle.

Line 159. Connect the mention of indirect ELISA with line 169 using the abbreviation in parentheses (iELISA).

Line 196. Regarding the positive and negative controls, it is necessary to mention briefly how the sera were standardized. For the positive control, its final reactivity (high, medium or low according to the cut-off value of the test) should be mentioned.

Line 208. Is this statement true for all sera included in the five sample groups?

Line 220. ... both commercial kits? Only one is identified in line 204.

Line 224. It is necessary to mention the post-immunization sampling scheme and briefly indicate what these serums were used for.

Line 225. Indicate in this section the analysis of results performed to determine the percentages of positive and negative agreement between the tests evaluated. Cohen's kappa index can also be applied to show the agreement observed in the 2x2 tables.

Line 243 which means N/P

Line 247. Cut off

Line 249. What were the serum diluents evaluated? Not indicated.

Line 251. Does this sentence refer to the serums listed in lines 114 to 118? No further mention is made of a positive standard serum.

Line 252-255. Rewrite to give clarity to the text.

Line 256. Which tests were applied to classify these samples as positive and negative?

Line 231. The results section will have to be revised and rewritten based on the modifications made in the Material and Methods section. The purpose will be to give congruence to the sequence of the text and clarity to the presentation of the results.

Line 266. What does PI mean?

Line 290. ... industry-standard HI assay. This is the first mention of this standard. It was not described in Material and Methods.

Line 295. This experiment was not identified in Material and Methods.

Line 391. Could the analytical sensitivity of the tests influence the results? This point should be discussed.

Line 319. Discussion

Line 34 mentions that the reference test is the inhibition of haemagglutination, but the discussion only compares the results of the two ELISAs. The purpose of each test selected for this study needs to be clarified. This section needs to be revised and rewritten according to the changes made in the previous sections, Materials and Methods and Results, to align all the information in the manuscript.

Reviewer #2 (Comments for the Author):

The manuscript titled "Development and application of an HRP-pet28a-NP and mAb-based cELISA for the detection of nucleoprotein antibodies of equine influenza virus", submitted by Yang et al., provides a study on the development of a cELISA to detect antibodies against equine influenza virus. The authors have expressed recombinant NP of equine influenza virus in mammalian cells and bacteria. The recombinant NP expressed in mammalian cells was used to develop hybridomas against the NP protein. The rNP protein expressed in bacteria was used in cELISA to bind with the positive serum sample. Authors tested the performance of their cELISA using 119 clinical samples and compared the results with Haemagglutination Inhibition (HI) assay and other commercially available cELISA kit. Authors showed agreement of cELISA results with the HI results. Authors also tested the efficiency of their cELISA to detect antibodies in horses immunized with inactivated vaccine. The study is very detailed, but the cELISA kits against NP have been developed since more than two decades ago, the manuscript lacks novelty. Although the importance of developing an indigenous cELISA kit, which is expensive otherwise to import and also suffers from logistic challenges, cannot be ignored.

Major comments:

1. NP and M gene being relatively highly conserved across influenza types are commonly used for typing of influenza viruses. Anti NP antibodies can be targeted to detect all influenza subtypes in type A. Authors have not emphasised the use of anti-NP antibodies for influenza type A detection.
2. Authors have not shown if rNP can be used as a universal antigen to detect all type A influenza subtypes including H5N1 and other low pathogenic viruses as NP can detect antibodies raised against diverse influenza subtypes (post infection)
3. Some useful information in the materials -methods is missing and which in my opinion should either be included or the tiny details should be uniformly removed throughout the manuscript (details in minor comments)
4. The discussion seems a bit lengthy and requires restructuring for easy understanding. A thorough edit can make it concise 1-2 pages which is easy to follow and understand.

Minor comments:

The current title does not clearly reflect the study's scope. Consider rephrasing to highlight the significance of nucleoprotein (NP) antibodies in the context of all influenza virus (EIV). Can this be used to detect other influenza subtypes which usually do not affect horses? I suggest it should be tested to broaden the applicability of the study.

The inclusion of "HRP-pET28a-NP" in the title seems unnecessary and could be omitted for clarity and conciseness.

Specific comments -

Abstract:

1. Line 25: Consider using "enzootic" instead of "endemic" for clarity in the veterinary context. In epidemiological contexts - endemic is used for human viruses and enzootic is used for veterinary viruses
2. Line 29: Please clarify which protein the purified monoclonal antibodies were generated against - is it NP?
3. Line 31: Specify whether the incubation was carried out sequentially or simultaneously with the serum.
4. Serum samples: Were these raised against a particular virus or protein? Please elaborate.
5. Line 37: Was the sero-surveillance study conducted using the same ELISA kit developed in this study?
6. Line 41: "Serological detection of EIV" is misleading - the study appears to report detection of EIV antibodies, not the virus itself. Consider rewording.

Introduction and Background:

1. Line 54: Typo - change "occur" to "occurs".
2. Line 92: NP is not a structural protein; this should be corrected.
3. Line 94: It would be valuable to mention that anti-NP antibodies can distinguish between influenza types A, B, and C, using type-specific ELISAs.
4. Line 99: cELISAs using RNP have been developed globally. Therefore, describing this method as a novel tool may be an overstatement. Please revise accordingly.

Materials and Methods:

1. Line 103: Provide detailed information on the plasmids used. Were they sourced from previous studies (relevant references)? Please elaborate on the details regarding the plasmids - how NP was cloned, what was the purification tag used etc.
2. What clade and lineage does the H3N8 strain used in the current study belong to?
3. Line 108: Clarify whether the EIV strain used belongs to the same lineage/clade as the NP donor strain.
4. Line 121: Indicate which EIV vaccine was used, along with clade information.
5. Line 126: What was the rationale for using a eukaryotic expression system for NP, given that NP does not generally require post-translational modifications?
6. Line 138: Why were two different versions of NP used? Clarify the purpose
7. Line 150: Please elaborate on the pVR1012-signal-PPT-H3N8XJ07NP-6xHis plasmid. Was this used as a DNA vaccine?
8. What was the rationale for plasmid immunization followed by rNP boosting? PI. explain
9. Line 158-160: Please expand on the hybridoma production process. Which cells were harvested, which myeloma line was used, and how was subcloning performed?
10. Line 170: Were proteins diluted in PBS or directly in the carbonate-bicarbonate buffer? Please clarify.
11. Line 174: If volumes of coating antigen and hybridoma supernatant are specified, also include the blocking buffer volume for reproducibility.
12. Line 179: Typo - remove the extra "in".
13. Line 180: How many additional washes were performed? Please specify.

14. Line 213: Describe how the 4HA units were calculated.
15. Line 216: Include the methodology for observing the inhibitory effect and how the HI titers were calculated.
16. Line 221: Provide clade information for the virus.
17. What concentration of BPL was used? Which adjuvant was used for emulsification?
18. Line 224: Were horses bled post-immunization for serum collection?
19. Line 239: Typo - Please use 1E5 followed by 1E10 for numerical consistency.
20. Lines 241-242: The sentence on coating mAbs and using HRP-pET28a-NP as the detection antigen is confusing - consider restructuring for clarity.
21. Line 243: Clarify what "N/P" refers to, as it is not defined earlier in the manuscript.

Results and Discussion:

1. Line 286: Provide the percentage agreement for IDvet cELISA.
2. Line 287: Define what is meant by "positive" and "negative" agreement - are these referring to sensitivity/specificity or another metric?
3. Line 331-333: Reiterate the type-specific nature of NP and its potential utility in distinguishing influenza A from other types.
4. Line 344: Explain how this approach reduces the amount of NP required
5. Line 351: Discuss why DNA prime-rNP boost might lead to stronger affinity of mAb
6. Line 360: The claim of a more efficient ELISA development strategy needs more justification - please elaborate on what makes this approach more efficient.
7. Line 364-365: These details may be redundant and could be removed to improve conciseness.
8. Line 385: NP detects type-specific antibodies; hence, NP-based cELISA could be useful for generic influenza A antibody detection. Testing against diverse sera (e.g., H5N1 in Mongolian horses) would strengthen this claim. PMID: 39661025 - Mongolian horses testing positive for H5N1
9. Latest references 2023 onwards are missing
10. It's imperative to know if the cELISA developed in the present study can detect antibodies in diverse species infected with a diverse range of influenza viruses. Copied some references below:
An ELISA for detection of antibodies against influenza A nucleoprotein in humans and various animal species - PubMed
A comparative evaluation of seven commercial human influenza virus antigen detection kits for the diagnosis of equine influenza - PubMed
Comments on 'Should the equine community be concerned about the emergence of the H5N1 subtype of highly pathogenic avian influenza in US cattle?' - PubMed

Reviewer #3 (Comments for the Author):

**Strong point :

- Highly relevant to equine disease surveillance.
- Use of a recombinant nucleoprotein expressed in a bacterial system: economical and reproducible method.
- Use of a specific monoclonal antibody: improves test specificity.
- Well-presented data supported by statistical tests
- Sound research methodology.

**Weaknesses or limitations :

- No cross-validation with other reference tests (e.g. HI, SRH).
- Potentially poor reproducibility if HRP coupling is poorly controlled.
- Low number of serums tested (sample size).
- Lack of analysis of cross-specificity with other equine or non-equine influenza viruses.
- Lack of data on the detection limit.

**Suggestions for improvement:

Add a comparative table with performance (sensitivity/specificity) versus a standard test.

Further discuss HRP-pET28a-NP protein stability (storage, degradation).

Add inter-batch / inter-day reproducibility analysis.

If possible, include a study of vaccinated vs. infected animal samples to assess serological discrimination.

Development and application of a HRP-pET28a-NP and mAb-based cELISA for the detection of nucleoprotein antibodies of equine influenza virus

After thorough examination and analysis of all the elements of this manuscript : this article presents the development of a competitive enzyme-linked immunosorbent assay (cELISA) using a recombinant nucleoprotein expressed via the pET28a vector and coupled to HRP. The test is based on the competitive interaction between serum antibodies from infected equids and an NP-specific monoclonal antibody. The test is designed to improve the sensitivity and specificity of serological diagnosis of equine influenza.

**Strong point:

- Highly relevant to equine disease surveillance.
- Use of a recombinant nucleoprotein expressed in a bacterial system: economical and reproducible method.
- Use of a specific monoclonal antibody: improves test specificity.
- Well-presented data supported by statistical tests
- Sound research methodology.

**Weaknesses or limitations:

- No cross-validation with other reference tests (e.g. HI, SRH).
- Potentially poor reproducibility if HRP coupling is poorly controlled.
- Low number of serums tested (sample size).
- Lack of analysis of cross-specificity with other equine or non-equine influenza viruses.
- Lack of data on the detection limit.

**Suggestions for improvement:

- Add a comparative table with performance (sensitivity/specificity) versus a standard test.

-Further discuss HRP-pET28a-NP protein stability (storage, degradation).

-Add inter-batch / inter-day reproducibility analysis.

-If possible, include a study of vaccinated vs. infected animal samples to assess serological discrimination.

In conclusion: this manuscript presents an original and well-structured work that proposes a promising diagnostic tool for equine influenza. A few methodological clarifications and additional analyses are required to improve its scope and reproducibility.

Recommendation: Minor revisions.

Response to the comments of the reviewer #1

Reviewer #1 (Comments for the Author):

The manuscript Development and application of a HRP-pET28a-NP and mAb-based cELISA for the detection of nucleoprotein antibodies of equine influenza virus was reviewed. The study included different experimental stages to develop, evaluate performance and apply a new serological test in population diagnosis.

After reading this manuscript I had identify interesting results. So, to contribute to the authors' improvement of this manuscript, I make the following comments.

Lines 33 and 34. OUR ESSAY? Which one? ... A commercial kit, which one?

Response:

We thank the reviewer for the question. We have corrected “To evaluate the performance of the cELISA, we tested 119 clinical samples using our assay, a commercial kit, and the hemagglutination inhibition (HI) assay as a reference” as “To assess the diagnostic performance of the NP-cELISA, we evaluated 119 clinical samples in parallel using the NP-cELISA, a commercially available competitive ELISA (ID.vet cELISA), and the hemagglutination inhibition (HI) assay as the reference standard“(track changes, line 33-36).

It is considered convenient to inform readers of the performance parameters of the Hemagglutination Inhibition test as it is the reference test with which the other tests are compared. In other words, the HI's validity as reference test will be strengthened by revealing its sensitivity and specificity.

Line 34. In this manuscript, it was omitted mention of the performance of the reference test inhibition of hemagglutination, with which the tests under evaluation are compared.

Response:

We thank the reviewer for the suggestion. We have added the description in the revised manuscript (track changes, line 93-94).

Among these, the HI assay is a prominent and commonly accepted method used to determine quantitative antibody titers for influenza virus (Bibby et al. 2022).

Line 101. This section should be rewritten by adding some paragraphs and rearranging existing paragraphs to give greater clarity to the manuscript.

The study design is multi-stage, as is shown in Table 1, indicated up to line 110. The flowchart created to realize the development and evaluation of serological tests is strongly recommended to be included at the beginning of this section of Material and Methods by a clear and brief text. Likewise, it is recommended that after describing the methodological design, the techniques used must be included in a structure that allows readers to identify which stage they correspond to and, in a short sentence, justify their use, that is, indicate what the authors intended to obtain when applying each technique. It is also highly recommended to make explicit in the section Serum samples, the characteristics of the samples that were used in each stage, since their mention in this manuscript is confusing and inconsistent for the reader: line 33=119, line 110=153, line 120 = 84, line 122=3,451.

Response:

We thank the reviewer for the suggestion. We have rewritten this section in the revised manuscript (track changes, line 135-156).

A total of 3817 sera were used in this study (Table 1). One positive control serum (HVRI-EIV-PS, HI titer 256×) was collected from a horse naturally infected with EIV (subtype H3N8, XJ lineage) in Xinjiang for the NP-cELISA optimization. To establish the Cut off value for the NP-cELISA, 153 serum samples collected from nationwide farms were used, with antibody status confirmed by both the hemagglutination inhibition (HI) test and ID.vet-cELISA (ID.Vet, Montpellier, France), including 60 positive and 93 negative samples. Eight antisera positive against other viruses or bacteria were selected for the analytical specificity test. Among them, antisera against equine herpesvirus type 1 (EHV-1), 4 (EHV-4), and equine arteritis virus (EAV) were purchased from National Veterinary Services Laboratories (NVSL, Trent, UK). Positive sera against *Salmonella Abortusequi* (*S. Abortusequi*), *Burkholderia mallei* (*B. mallei*), *Theileria equi* (*T. equi*), *Babesia caballi* (*B. caballi*), avian influenza virus (AIV, H7N4, H9N2), were purchased from National Engineering Center for Veterinary Biological (NECVB, Harbin, China). To further

evaluate the detection performance of the NP-cELISA, 119 clinical serum samples and 84 horse serum samples immunized with EIV inactivated antigen (H3N8 subtype, XJ lineage) were collected and tested using this method. Finally, a total of 3,451 equine serum samples collected from 2021 to 2023 across multiple Chinese provinces were analyzed using the NP-cELISA to monitor the seroprevalence of equine influenza (EI) antibodies in China. All serum samples were aseptically collected and transported to the laboratory following standard procedures and stored at -20 °C until further analysis.

Line 107. Five groups of samples are identified. You could indicate this at the beginning of this section and perhaps number them to make it clearer to the reader. It is also important to clarify whether ALL samples were processed using the three techniques, iELISA, cELISA, and haemagglutination inhibition.

Response:

We thank the reviewer for the suggestion. The detection techniques used for serum samples have been clearly defined in the manuscript and the table 1.(track changes, line 135-156, Line 669).

Line 108. Considering the value of this serum in the present study, it will be convenient to mention the characteristics of this serum (origin, tests used for classification as positive, storage, etc.).

Response:

We thank the reviewer for the suggestion. We have added the corresponding information in the revised manuscript (track changes, line 135-137).

A total of 3817 sera were used in this study (Table 1). One positive control serum (HVRI-EIV-PS, HI titer 256×) was collected from a horse naturally infected with EIV (subtype H3N8, XJ lineage) in Xinjiang for the NP-cELISA optimization.

Line 113. The ID.vet test corresponds to an ELISA test (line 204). Since the first mention in the abstract and this line 113, I recommend indicating that it is an ELISA test.

Response:

We thank the reviewer for the suggestion. We have added the information in the

revised manuscript (track changes, line 33-36).

To assess the diagnostic performance of the NP-cELISA, we evaluated 119 clinical samples in parallel using the NP-cELISA, a commercially available competitive ELISA (**ID.vet cELISA**), and the hemagglutination inhibition (HI) assay as the reference standard.

Lines 125, 149, 169, 128. I suggested writing the full name and abbreviations in parentheses in the subtitle.

Response:

We thank the reviewer for the suggestion. We have added information in the revised manuscript.

2.3 Preparation of the nucleoprotein (NP) (track changes, line 157).

2.4 Immunization of mice and monoclonal antibody (mAb) production (track changes, line 182).

2.5 indirect ELISA (iELISA) (track changes, line 205).

2.7 Hemagglutination Inhibition Test (HI)(track changes, line 242).

Line 159. Connect the mention of indirect ELISA with line 169 using the abbreviation in parentheses (iELISA).

Response:

We thank the reviewer for the suggestion. We have added information in the revised manuscript (track changes, line 205).

2.5 indirect ELISA (iELISA)

Line 208. Is this statement true for all sera included in the five sample groups?

Response:

We thank the reviewer for the suggestion. We have added the description in the revised manuscript (track changes, line 245-247).

In HI test, all sera samples were beforehand treated (16 h) with receptor destroying enzyme (RDE) (Sigma C8772), at a 1:4 dilution, followed by heat inactivation at 56 °C for 30 minutes.

Line 220. .both commercial kits? Only one is identified in line 204.

Response:

We thank the reviewer for the question. We have corrected "both commercial kits" as "ID.vet cELISA" in the revised manuscript. (track changes, line 257-258).

Four horses confirmed to be negative for equine influenza antibodies, as determined by both **ID.vet cELISA** and the HI test, were selected for immunization. Line 224. It is necessary to mention the post-immunization sampling scheme and briefly indicate what these serums were used for.

Response:

We thank the reviewer for the suggestion. We have added description in the revised manuscript (track changes, line 262-266).

Serum samples was collected from horses before immunization; from day 5 after immunization, with daily collections on days 5-11; every 3 days on days 11-38; and every 2 weeks thereafter. A total of 84 serum (21×4) collections were made in this experiment. Seroconversion was then evaluated using the NP-cELISA and HI assays. Line 225. Indicate in this section the analysis of results performed to determine the percentages of positive and negative agreement between the tests evaluated. Cohen's kappa index can also be applied to show the agreement observed in the 2x2 tables.

Response:

We thank the reviewer for the suggestion. We have added the Cohen's kappa index in the revised manuscript (track changes, Table 2,3, Line 676, 716).

Line 243 which means N/P

Response:

We thank the reviewer for the question. We have corrected "N/P" as " $OD_{\text{Negative}}/OD_{\text{Positive}}$ (N/P)" in the revised manuscript. (track changes, line 304).

Line 247. Cut off

Response:

We thank the reviewer for the suggestion. We have corrected "cut-off" as "Cut off value" in the revised manuscript. (track changes, line 3256, line 328, line 330).

Line 249. What were the serum diluents evaluated? Not indicated.

Response:

We thank the reviewer for the question. We have added the serum diluents in the revised manuscript (track changes 313).(Solution I, II, III, (PR-SS-001 to 003), InnoReagents, Hubei, China).

Line 251. Does this sentence refer to the serums listed in lines 114 to 118? No further mention is made of a positive standard serum.

Response:

We thank the reviewer for the question. Yes, this sentence refers to the serums listed in lines 114 to 118. We have corrected "standard positive serum" as "EIV positive serum" in the revised manuscript. (track changes, line 314-315).

Line 252-255. Rewrite to give clarity to the text.

Response:

We thank the reviewer for the suggestion. We have rewritten it in the revised manuscript (track changes 318-322).

Subsequently, the absorbance values at 450 nm (OD450) of the diluted serum samples were measured and recorded. Then, the ratio of the absorbance of the N/P value was calculated to evaluate the performance of each dilution condition. The experimental results showed that the highest N/P value was obtained when Solution I was used as the diluent and the serum dilution was 8.

Line 256. Which tests were applied to classify these samples as positive and negative?

Response:

We thank the reviewer for the question. The 153 sera verified by HI and ID. vet-cELISA (93 naive sera and 60 infected sera) were further evaluated using the newly developed NP-cELISA to determine the cut off value, sensitivity, and specificity of the assay.(track changes 323-325).

Line 231. The results section will have to be revised and rewritten based on the modifications made in the Material and Methods section. The purpose will be to give congruence to the sequence of the text and clarity to the presentation of the results.

Line 266. What does PI mean?

Response:

We thank the reviewer for the question. We have added the full name of PI (Percent inhibition) in the revised manuscript (track changes 325).

Line 290. ...industry-standard HI assay. This is the first mention of this standard. It was not described in Material and Methods.

Response:

We thank the reviewer for the suggestion. We have rewritten it in the revised manuscript (track changes 355-357).

To investigate the discordance between the serological results of the NP cELISA and ID.vet cELISA, the WOAHA-recommended HI assay was used for further confirmation.

Line 295. This experiment was not identified in Material and Methods.

Response:

We thank the reviewer for the question. We have added description in the revised manuscript (track changes 257-266).

Four horses confirmed to be negative for equine influenza antibodies, as determined by both ID.vet cELISA and the HI test, were selected for immunization. The Chinese strain of EIV, A/equine/XinjiangFuyun/3/07 (subtype H3N8, XJ lineage), was inactivated with β -propiolactone (0.25% v/v) and mixed with an adjuvant (MONTANIDE ISA35) at a 1:1 volume ratio. Each horse was immunized with 4 mL of the inactivated virus-adjuvant mixture. Serum samples were collected from horses before immunization; from day 5 after immunization, with daily collections on days 5-11; every 3 days on days 11-38; and every 2 weeks thereafter. A total of 84 serum (21 \times 4) collections were made in this experiment. Seroconversion was then evaluated using the NP-cELISA and HI assays.

Line 391. Could the analytical sensitivity of the tests influence the results? This point should be discussed.

Response:

We thank the reviewer for the question. We have added the discussion in the revised manuscript (track changes 459-464).

The NP-cELISA detected seroconversion earlier than HI in 3 out of 4 horses (Fig. 6), likely due to differences in the targeted antigens: NP for the NP-cELISA and HA for HI. Variations in antibody production timing and levels, as well as individual immune responses, may also contribute to this discrepancy (Bahadoran et al. 2016; McFarlane 2016). In addition, the difference in analytical sensitivity between the two tests can also affect the results.

Line 319. Discussion

Line 34 mentions that the reference test is the inhibition of haemagglutination, but the discussion only compares the results of the two ELISAs. The purpose of each test selected for this study needs to be clarified. This section needs to be revised and rewritten according to the changes made in the previous sections, Materials and Methods and Results, to align all the information in the manuscript.

Response:

We thank the reviewer for the suggestion. We have added discussion in the revised manuscript (track changes lines 452-457).

Among 119 clinical samples, the positive detection rates were comparable between NP-cELISA (69.7%, 83/119) and HI assay (72.3%, 86/119), both significantly higher than ID.vet-cELISA (55.5%, 66/119). Using the WOAHA-recommended HI assay as the reference standard, the analysis revealed superior concordance between NP-cELISA and HI (87.4%) compared to that between ID.vet cELISA and HI (78.2%) (Table 3).

Reviewer #2 (Comments for the Author):

The manuscript titled "Development and application of an HRP-pet28a-NP and mAb-based cELISA for the detection of nucleoprotein antibodies of equine influenza virus", submitted by Yang et al., provides a study on the development of a cELISA to detect antibodies against equine influenza virus. The authors have expressed

recombinant NP of equine influenza virus in mammalian cells and bacteria. The recombinant NP expressed in mammalian cells was used to develop hybridomas against the NP protein. The rNP protein expressed in bacteria was used in cELISA to bind with the positive serum sample. Authors tested the performance of their cELISA using 119 clinical samples and compared the results with Haemagglutination Inhibition (HI) assay and other commercially available cELISA kit. Authors showed agreement of cELISA results with the HI results. Authors also tested the efficiency of their cELISA to detect antibodies in horses immunized with inactivated vaccine. The study is very detailed, but the cELISA kits against NP have been developed since more than two decades ago, the manuscript lacks novelty. Although the importance of developing an indigenesous cELISA kit, which is expensive otherwise to import and also suffers from logistic challenges, cannot be ignored.

Major comments:

1. NP and M gene being relatively highly conserved across influenza types are commonly used for typing of influenza viruses. Anti NP antibodies can be targeted to detect all influenza subtypes in type A. Authors have not emphasised the use of anti-NP antibodies for influenza type A detection.

Response:

We thank the reviewer for the suggestion. We have added the discussion in the revised manuscript (track changes435-446).

To verify whether the NP-cELISA developed can detect antibodies raised after infection by diverse influenza subtypes in different species. Firstly, multiple sequence alignment analysis showed that the amino acid sequences of different influenza A virus NP proteins were highly conserved (Fig. S1). We further verified the cross-reactivity of 1E10 with other influenza A viruses and found that this monoclonal antibody could react with all the influenza A viruses NP selected in this study (Fig. 4). The cross-specificity of the developed NP-cELISA was further validated using a series serum against EIV (H3N8) and AIV (H7N4, H9N2). 1E10-based NP-cELISA might be used for the detection of antibodies against different subtypes of influenza A

viruses. The broader applicability of the NP-cELISA for influenza type A detection across multiple subtypes was needed further validated in future study, due to the unavailability of clinical samples of other species.

2. Authors have not shown if rNP can be used as a universal antigen to detect all type A influenza subtypes including H5N1 and other low pathogenic viruses as NP can detect antibodies raised against diverse influenza subtypes (post infection).

Response:

We thank the reviewer for the question. We have added it in the revised manuscript (track changes 288-292).

The amino acid sequences of the NP proteins from different EIV strains were aligned with homologous sequences from other influenza A virus strains. Amino acid sequence analysis revealed that the NP proteins were highly conserved, showing 91.4–99.4% identity, and could serve as a universal antigen for detecting all influenza A subtypes (Fig. S1).

3. Some useful information in the materials -methods is missing and which in my opinion should either be included or the tiny details should be uniformly removed throughout the manuscript (details in minor comments)

Response:

We thank the reviewer for the suggestion. We have corrected it in the revised manuscript.

4. The discussion seems a bit lengthy and requires restructuring for easy understanding. A thorough edit can make it concise 1-2 pages which is easy to follow and understand.

Response:

We thank the reviewer for the suggestion. In this study, while streamlining redundant portions of the original discussion, we have also expanded new dimensions of analysis in response to the other reviewers' suggestions.

Minor comments:

The current title does not clearly reflect the study's scope. Consider rephrasing to

highlight the significance of nucleoprotein (NP) antibodies in the context of all influenza viruses (EIV). Can this be used to detect other influenza subtypes which usually donot affect horses? I suggest it should be tested to broaden the applicability of the study.

Response:

We thank the reviewer for the suggestion. We have added the discussion in the revised manuscript (track changes 435-446).

To verify whether the NP-cELISA developed can detect antibodies raised after infection by diverse influenza subtypes in different species. Firstly, multiple sequence alignment analysis showed that the amino acid sequences of different influenza A virus NP proteins were highly conserved (Fig. S1). We further verified the cross-reactivity of 1E10 with other influenza A viruses and found that this monoclonal antibody could react with all the influenza A viruses NP selected in this study (Fig. 4). The cross-specificity of the developed NP-cELISA was further validated using a series serum against EIV (H3N8) and AIV (H7N4, H9N2). 1E10-based NP-cELISA might be used for the detection of antibodies against different subtypes of influenza A viruses. The broader applicability of the NP-cELISA for influenza type A detection across multiple subtypes was needed further validated in future study, due to the unavailability of clinical samples of other species.

The inclusion of "HRP-pET28a-NP" in the title seems unnecessary and could be omitted for clarity and conciseness.

Response:

We thank the reviewer for the suggestion. We have deleted it in the revised manuscript.

Development and application of a NP-cELISA for the detection of nucleoprotein antibodies of equine influenza virus

Specific comments -

Abstract:

1. Line 25: Consider using "enzootic" instead of "endemic" for clarity in the veterinary context. In epidemiological contexts - endemic is used for human viruses and enzootic is used for veterinary viruses

Response:

We thank the reviewer for the suggestion. We have corrected it in the revised manuscript (track changes line 25).

Equine influenza (EI), caused by the equine influenza virus (EIV), is an acute respiratory disease that has become **enzootic** worldwide, resulting in frequent outbreaks and substantial economic losses within the equine industry

2. Line 29: Please clarify which protein the purified monoclonal antibodies were generated against - is it NP?

Response:

We thank the reviewer for the question. We have corrected it in the revised manuscript (track changes line 28-31).

The assay was designed by coating plates with purified monoclonal antibodies (mAbs) against the NP protein, followed by simultaneous incubation of the test serum samples and HRP-NP antigen in a competitive binding reaction

3. Line 31: Specify whether the incubation was carried out sequentially or simultaneously with the serum.

Response:

We thank the reviewer for the question. We have corrected this sentence as “The assay was designed by coating plates with purified monoclonal antibodies (mAbs) against the NP protein, followed by simultaneous incubation of the test serum samples and HRP-NP antigen in a competitive binding reaction”(track changes line 28-31)

4. Serum samples: Were these raised against a particular virus or protein? Please elaborate.

Response:

We thank the reviewer for the question. Serum samples refer to the general specimens to be tested, and their specific information has been described in Table 1 and the Materials and Methods section.

5. Line 37: Was the sero-surveillance study conducted using the same ELISA kit developed in this study?

Response:

We thank the reviewer for the question. We have added “using the developed NP-cELISA” in the revised manuscript (track changes line 39).

6. Line 41: "Serological detection of EIV" is misleading - the study appears to report detection of EIV antibodies, not the virus itself. Consider rewording.

Response:

We thank the reviewer for the suggestion. We have corrected this sentence as “Serological detection of EI”.(track changes line 43).

In conclusion, the NP-cELISA developed in this study demonstrates significant potential as a reliable and efficient diagnostic tool for the serological detection of EI, with broad applicability in various settings

Introduction and Background:

1. Line 54: Typo - change "occur" to "occurs".

Response:

We thank the reviewer for the suggestion. We have corrected it in the revised manuscript (track changes line 68).

2. Line 92: NP is not a structural protein; this should be corrected.

Response:

We thank the reviewer for the suggestion. We have corrected this sentence as the follows. The nucleoprotein (NP) is the most abundant protein encoded by segment 5 of the influenza A virus (IAV) genome. (track changes line 108-109).

3. Line 94: It would be valuable to mention that anti-NP antibodies can distinguish between influenza types A, B, and C, using type-specific ELISAs.

Response:

We thank the reviewer for the suggestion. We have added this sentence in the

revised manuscript (track changes line 108-112).

The nucleoprotein (NP) is the most abundant protein encoded by segment 5 of the influenza A virus (IAV) genome. It consists of 498 amino acids and has a molecular weight of 56 kDa (Hu et al. 2017). It would be valuable to mention that anti-NP antibodies can distinguish between influenza types A, B, C and D, using type-specific ELISAs

4. Line 99: cELISAs using NP have been developed globally. Therefore, describing this method as a novel tool may be an overstatement. Please revise accordingly.

Response:

We thank the reviewer for the suggestion. We have corrected “novel” as “important” in the revised manuscript (track changes line 117).

Materials and Methods:

1. Line 103: Provide detailed information on the plasmids used. Were they sourced from previous studies (relevant references)? Please elaborate on the details regarding the plasmids - how NP was cloned, what was the purification tag used etc.

Response:

We thank the reviewer for the suggestion. We have added description in the revised manuscript (track changes line 121-130).

A total of 16 amino acid sequences of the NP proteins from influenza types A isolated from equine, chicken, duck, human, swine, and canine were aligned using the Megalign software (DNASTar, USA). Then, the EIV NP gene (Genbank: EU794544.1, subtype H3N8, XJ lineage) was amplified and cloned into pCAGGS, pET28a, and pVR1012 vector, respectively. The recombinant plasmids pCAGGS-GST-H3N8_{XJ07}-NP, pET28a-His-NP, and pVR1012-signalPPT-H3N8_{XJ07}-NP-6×His were then transformed into DH5α chemically competent cells (Tiangen, China) and purified with FastPure Gel DNA Extraction Mini Kit (Vazyme, China) according to the manufacturer’s instructions. NP genes of influenza A from other species were directly synthesized into pCAGGS vector and stored in the laboratory.

2. What clade and lineage does the H3N8 strain used in the current study belong to?

Response:

We thank the reviewer for the question. We have added the lineage of the H3N8 strain (subtype H3N8, XJ lineage) in the revised manuscript (track changes lin130-133).

The EIV (A/equine/XinjiangFuyun/3/07) used in this study was derived from a Chinese epidemic strain (subtype H3N8, XJ lineage) preserved in the laboratory and inactivated with β propiolactone (0.25%, v/v).

3. Line 108: Clarify whether the EIV strain used belongs to the same lineage/clade as the NP donor strain.

Response:

We thank the reviewer for the question. We have added description in the revised manuscript (track changes line135-137).

A total of 3817 sera were used in this study (Table 1). One positive control serum (HVRI-EIV-PS, HI titer 256 \times) was collected from a horse naturally infected with EIV (**subtype H3N8, XJ lineage**) in Xinjiang for the NP-cELISA optimization.

4. Line 121: Indicate which EIV vaccine was used, along with clade information.

Response:

We thank the reviewer for the suggestion. We have corrected this sentence as follows. The EIV (A/equine/XinjiangFuyun/3/07) used in this study was derived from a Chinese epidemic strain (subtype H3N8, XJ lineage) preserved in the laboratory and inactivated with β propiolactone (0.25%, v/v).(track changes line130-133).

5. Line 126: What was the rationale for using a eukaryotic expression system for NP, given that NP does not generally require post-translational modifications?

Response:

We thank the reviewer for the question. We have added discussion in the revised manuscript (track changes line 412-414).

The eukaryotic expression system, which produces proteins closer to the true

conformation of NP proteins than the prokaryotic system, reduces the amount of NP required for immunization compared to conventional prokaryotic expression methods.

6. Line 138: Why were two different versions of NP used? Clarify the purpose

Response:

We thank the reviewer for the question. Prokaryotic purification of antigens is less costly and more productive than eukaryotic purification, making it suitable for use during high-throughput assays.

7. Line 150: Please elaborate on the pVR1012-signal-PPT-H3N8XJ07NP-6xHis plasmid. Was this used as a DNA vaccine?

We thank the reviewer for the question. The plasmid (pVR1012-signal-PPT-H3N8XJ07NP-6xHis) was indeed used in immunization experiments. However, it is important to specifically clarify that it was not employed as a vaccine, but rather as a research tool to elicit immune responses (such as antibody production).

8. What was the rationale for plasmid immunization followed by rNP boosting? Pl. explain

We thank the reviewer for the question. The mAbs obtained through this immunization approach exhibited higher specificity and stronger binding affinity to the EIV NP than those produced by other methods (Schardt et al. 2021) (track changes line 414-420).

Plasmid immunization combined with protein immunization strategy reduce the use of NP proteins (Used only once during immunization) compared to conventional methods. DNA primary immunization promotes B-cell affinity maturation by prolonging the germinal center response through sustained antigen expression, while eukaryotic protein enhancement selectively expands high-affinity B-cell clones through high concentrations of natural antigens, both synergistically driving efficient antibody affinity maturation.

9. Line 158-160: Please expand on the hybridoma production process. Which cells

were harvested, which myeloma line was used, and how was subcloning performed?

Response:

We thank the reviewer for the suggestion. We have added the description in the revised manuscript (track changes line191-196).

Splenocytes were taken and fused with myeloma cells SP/20 and then cultured in 96-well plates for 5-7 days. Supernatants from growing hybridoma cells were screened using an indirect ELISA (iELISA) for reactivity to the NP protein. Positive hybridoma clones were diluted according to 1/well and then subcloned in 96-well culture plates until a positive monoclonal cell line was obtained.

10. Line 170: Were proteins diluted in PBS or directly in the carbonate-bicarbonate buffer? Please clarify.

Response:

We thank the reviewer for the suggestion. Proteins were diluted in PBS in this study. (track changes line 207).

11. Line 174: If volumes of coating antigen and hybridoma supernatant are specified, also include the blocking buffer volume for reproducibility.

Response:

We thank the reviewer for the suggestion. We have added the blocking buffer volume (**200 μ L/well**) in the revised manuscript (track changes line 211).

12. Line 179: Typo - remove the extra "in".

Response:

We thank the reviewer for the suggestion. We have removed the extra "in" in the revised manuscript

13. Line 180: How many additional washes were performed? Please specify.

Response:

We thank the reviewer for the suggestion. We have added "Following three times washing" in the revised manuscript (track changes line 216-217).

14. Line 213: Describe how the 4HA units were calculated.

Response:

We thank the reviewer for the suggestion. We have added the description in the revised manuscript (track changes line 243-245).

The agglutination potency of the viral hemagglutinin (HA) protein is first determined by the hemagglutination assay (HA assay), and then the antigen is diluted to a concentration of 1/4 potency, i.e. 4HA.

15. Line 216: Include the methodology for observing the inhibitory effect and how the HI titers were calculated.

Response:

We thank the reviewer for the suggestion. We have added the description in the revised manuscript (track changes line 253-255).

The highest serum dilution that exhibits complete hemagglutination inhibition (i.e., red blood cells settle at the bottom of the well as a cell pellet) is considered the HI titer of the serum.

16. Line 221: Provide clade information for the virus.

Response:

We thank the reviewer for the suggestion. We have added “subtype H3N8, XJ lineage” in the revised manuscript (track changes line 259).

17. What concentration of BPL was used? Which adjuvant was used for emulsification?

Response:

We thank the reviewer for the question. We have added the description in the revised manuscript (track changes line 259-261).

The Chinese strain of EIV, A/equine/XinjiangFuyun/3/07 (subtype H3N8, XJ lineage), was inactivated with β -propiolactone (0.25% v/v) and mixed with an adjuvant (MONTANIDE ISA35) at a 1:1 volume ratio.

18. Line 224: Were horses bled post-immunization for serum collection?

Response:

We thank the reviewer for the question. We have added the description in the revised manuscript (track changes, line 262-264).

Serum samples was collected from horses before immunization; from day 5 after immunization, with daily collections on days 5-11; every 3 days on days 11-38; and every 2 weeks thereafter. A total of 84 serum (21×4) collections were made in this experiment. Seroconversion was then evaluated using the NP-cELISA and HI assays.

19. Line 239: Typo - Please use 1E5 followed by 1E10 for numerical consistency.

Response:

We thank the reviewer for the suggestion. We have corrected it in the revised manuscript (track changes line 299).

20. Lines 241-242: The sentence on coating mAbs and using HRP-pET28a-NP as the detection antigen is confusing - consider restructuring for clarity.

Response:

We thank the reviewer for the suggestion. We have corrected it in the revised manuscript (track changes line 300-303).

To identify the mAb with the highest competitive activity for cELISA, four mAbs were coated onto ELISA plates at varying dilutions, followed by simultaneous incubation of the test serum samples and HRP-NP antigen in a competitive binding reaction.

21. Line 243: Clarify what "N/P" refers to, as it is not defined earlier in the manuscript.

Response:

We thank the reviewer for the suggestion. We thank the reviewer for the question. We have corrected "N/P" as " $OD_{\text{Negative}}/OD_{\text{Positive}}$ (N/P)" in the revised manuscript. (track changes, line 304).

Results and Discussion:

1. Line 286: Provide the percentage agreement for IDvet cELISA.

Response:

We thank the reviewer for the suggestion. We have added the percentage agreement for IDvet cELISA in the revised manuscript (track changes, line 352-355).

A total of 119 serum samples were tested using both assays. Compared to ID.vet-cELISA, the newly developed NP-cELISA demonstrated a sensitivity of 100% (66/66) and a specificity of 67.9% (36/53) (Table 2). The overall agreement between the two assays was 85.7% ((66+36)/119).

2. Line 287: Define what is meant by "positive" and "negative" agreement - are these referring to sensitivity/specificity or another metric?

Response:

We thank the reviewer for the question. We have corrected "positive/ negative" as "sensitivity/specificity" in the revised manuscript (track changes, line 352-354).

Compared to ID.vet-cELISA, the newly developed NP-cELISA has a sensitivity of 100% (66/66) and a specificity of 67.9% (36/ (36+17)).

3. Line 331-333: Reiterate the type-specific nature of NP and its potential utility in distinguishing influenza A from other types.

Response:

We thank the reviewer for the suggestion. We have added the description in the revised manuscript (track changes, line 396-398).

However, nucleoprotein (NP) is increasingly considered an alternative target for antibody detection due to its high conservation and ability to distinguish influenza A from other types(Hu et al. 2017; Rak et al. 2023).

4. Line 344: Explain how this approach reduces the amount of NP required

Response:

We thank the reviewer for the suggestion. We have added the description in the revised manuscript (track changes, line 414-416)

Plasmid immunization combined with protein immunization strategy reduce the use of NP proteins (Used only once during immunization) compared to conventional methods.

5. Line 351: Discuss why DNA prime-rNP boost might lead to stronger affinity of mAb

Response:

We thank the reviewer for the suggestion. We have added the discussion in the revised manuscript (track changes, line 416-420)

DNA primary immunization promotes B-cell affinity maturation by prolonging the germinal center response through sustained antigen expression, while eukaryotic protein enhancement selectively expands high-affinity B-cell clones through high concentrations of natural antigens, both synergistically driving efficient antibody affinity maturation.

6. Line 360: The claim of a more efficient ELISA development strategy needs more justification - please elaborate on what makes this approach more efficient.

Response:

We thank the reviewer for the suggestion. We have added the discussion in the revised manuscript (track changes, line 427-431)

Unlike conventional solid-phase ELISAs that use labeled mAbs, this method relies on liquid-phase interactions between the serum sample and HRP-pET28a-NP, which is more favorable for antigen-antibody reactions. This approach may provide a more efficient ELISA development strategy at reaction times (one-step reaction).

7. Line 364-365: These details may be redundant and could be removed to improve conciseness.

Response:

We thank the reviewer for the suggestion. We have deleted these details in the revised manuscript

8. Line 385: NP detects type-specific antibodies; hence, NP-based cELISA could be useful for generic influenza A antibody detection. Testing against diverse sera (e.g., H5N1 in Mongolian horses) would strengthen this claim. PMID: 39661025 - Mongolian horses testing positive for H5N1.

Response:

We thank the reviewer for the suggestion. The cross-specificity of the developed cELISA was further validated using a series serum against EIV (H3N8) and AIV (H7N4, H9N2). At present, we are unable to obtain other positive sera for influenza. Further research will be conducted pending the availability of serum samples.(track changes, line 441-442)

9. Latest references 2023 onwards are missing

Response:

We thank the reviewer for the suggestion. We have added the new references in the revised manuscript.

10. It's imperative to know if the cELISA developed in the present study can detect antibodies in diverse species infected with a diverse range of influenza viruses.

Copied some references below:

An ELISA for detection of antibodies against influenza A nucleoprotein in humans and various animal species - PubMed

A comparative evaluation of seven commercial human influenza virus antigen detection kits for the diagnosis of equine influenza - PubMed

Comments on 'Should the equine community be concerned about the emergence of the H5N1 subtype of highly pathogenic avian influenza in US cattle?' - PubMed

Response:

We thank the reviewer for the suggestion. To verify whether the NP-cELISA developed can detect antibodies raised after infection by diverse influenza subtypes in different species. Firstly, multiple sequence alignment analysis showed that the amino acid sequences of different influenza A virus NP proteins were highly conserved (Fig. S1). We further verified the cross-reactivity of 1E10 with other influenza A viruses and found that this monoclonal antibody could react with all the influenza A viruses NP selected in this study (Fig. 4). The cross-specificity of the developed NP-cELISA was further validated using a series serum against EIV (H3N8) and AIV (H7N4,

H9N2). 1E10-based NP-cELISA might be used for the detection of antibodies against different subtypes of influenza A viruses. The broader applicability of the NP-cELISA for influenza type A detection across multiple subtypes was needed further validated in future study, due to the unavailability of clinical samples of other species. (track changes, line435-446)

Reviewer #3 (Comments for the Author):

****Strong point :**

- Highly relevant to equine disease surveillance.
- Use of a recombinant nucleoprotein expressed in a bacterial system: economical and reproducible method.
- Use of a specific monoclonal antibody: improves test specificity.
- Well-presented data supported by statistical tests
- Sound research methodology.

****Weaknesses or limitations :**

- No cross-validation with other reference tests (e.g. HI, SRH).

Response:

We thank the reviewer for the suggestion. We have used the WOAHA-recommended HI assay as the reference standard in this manuscript.

- Potentially poor reproducibility if HRP coupling is poorly controlled.

Response:

We thank the reviewer for the suggestion. We agree that potentially poor reproducibility if HRP coupling is poorly controlled. To further validate inter-batch reproducibility, the positive serum sample was tested with three replicates in three independent batches of HRP-NP conjugate, demonstrating comparable sensitivity across all batches (Fig. 5D).(track changes, line347-349).

- Low number of serums tested (sample size).

Response:

We thank the reviewer for the suggestion. The samples we used in the method evaluation were indeed few. We will further evaluate this method in the subsequent study.

-Lack of analysis of cross-specificity with other equine or non-equine influenza viruses.

Response:

We thank the reviewer for the suggestion. To verify whether the NP-cELISA developed can detect antibodies raised after infection by diverse influenza subtypes in different species. Firstly, multiple sequence alignment analysis showed that the amino acid sequences of different influenza A virus NP proteins were highly conserved (Fig. S1). We further verified the cross-reactivity of 1E10 with other influenza A viruses and found that this monoclonal antibody could react with all the influenza A viruses NP selected in this study (Fig. 4). The cross-specificity of the developed NP-cELISA was further validated using a series serum against EIV (H3N8) and AIV (H7N4, H9N2). 1E10-based NP-cELISA might be used for the detection of antibodies against different subtypes of influenza A viruses. The broader applicability of the NP-cELISA for influenza type A detection across multiple subtypes was needed further validated in future study, due to the unavailability of clinical samples of other species..(track changes, line435-446).

****Suggestions for improvement:**

Add a comparative table with performance (sensitivity/specificity) versus a standard test.

Response:

We thank the reviewer for the suggestion. We have added a comparative table with performance (sensitivity/specificity) versus the WOAHA-recommended HI assay as the reference standard in this manuscript. (track changes, line716-718).

Further discuss HRP-pET28a-NP protein stability (storage, degradation).

Response:

We thank the reviewer for the suggestion. We have added the description in the revised manuscript (track changes, line 448-452).

The stability of the HRP-NP protein and its labeling efficiency will affect both the repeatability and sensitivity of the assay results. Inter-batch reproducibility was validated by testing the positive serum in triplicate across three independent HRP-NP conjugate batches, showing consistent sensitivity (256×, equal to HI titer).

Add inter-batch / inter-day reproducibility analysis.

Response:

We thank the reviewer for the suggestion. To further validate inter-batch reproducibility, the positive serum sample was tested with three replicates in three independent batches of HRP-NP conjugate, demonstrating comparable sensitivity across all batches (Fig. 5D). (track changes, line 347-349).

If possible, include a study of vaccinated vs. infected animal samples to assess serological discrimination.

Response:

We thank the reviewer for the suggestion. We will further a study for vaccinated vs. infected animal samples to assess serological discrimination.

Bahadoran A, Lee SH, Wang SM, Manikam R, Rajarajeswaran J, Raju CS, Sekaran SD (2016) Immune Responses to Influenza Virus and Its Correlation to Age and Inherited Factors. *Frontiers in Microbiology* 7 doi:10.3389/fmicb.2016.01841

Bibby DC, Savanovic M, Zhang J, Torelli A, Jeeninga RE, Gagnon L, Harris SL (2022) Interlaboratory Reproducibility of Standardized Hemagglutination Inhibition Assays. *mSphere* 7(1):e0095321 doi:10.1128/msphere.00953-21

Hu Y, Sneyd H, Dekant R, Wang J (2017) Influenza A Virus Nucleoprotein: A Highly Conserved Multi-Functional Viral Protein as a Hot Antiviral Drug Target. *Current Topics in Medicinal Chemistry* 17(20) doi:10.2174/1568026617666170224122508

McFarlane D (2016) Immune Dysfunction in Aged Horses. *Veterinary Clinics of North America: Equine Practice* 32(2):333-341
doi:10.1016/j.cveq.2016.04.009

Rak A, Isakova-Sivak I, Rudenko L (2023) Nucleoprotein as a Promising Antigen for Broadly Protective Influenza Vaccines. *Vaccines* 11(12)
doi:10.3390/vaccines11121747

Schardt JS, Pornnoppadol G, Desai AA, Park KS, Zupancic JM, Makowski EK, Smith MD, Chen H, Garcia de Mattos Barbosa M, Cascalho M, Lanigan TM, Moon JJ, Tessier PM (2021) Discovery and characterization of high-affinity, potent SARS-CoV-2 neutralizing antibodies via single B cell screening. *Scientific Reports* 11(1) doi:10.1038/s41598-021-99401-x

Re: Spectrum00939-25R1 (Development and application of a NP-cELISA for the detection of nucleoprotein antibodies of equine influenza virus)

Dear Prof. Xiaojun Wang:

Thank you for the privilege of reviewing your work. I am pleased to inform you that your manuscript has been editorially accepted for publication. However, there are a few additional questions from the reviewers that need to be answered before the final decision. Once these are completed, please return your submission within 5 days so that I can move your paper forward to acceptance.

If you cannot complete the modification within this time period, please contact me. If you do not wish to modify the manuscript and prefer to submit it to another journal, notify me immediately so that the manuscript may be formally withdrawn from consideration by Spectrum.

Revision Guidelines

Sincerely,
Yunyu Chen
Editor
Microbiology Spectrum

Reviewer #1 (Comments for the Author):

Table 2. What is the HI technique result for the 17 samples with discordant ELISA test results? Could you include it in Section 3.4?

Response to the comments of the reviewer #1

Reviewer #1 (Comments for the Author):

Table 2. What is the HI technique result for the 17 samples with discordant ELISA test results? Could you include it in Section 3.4?

Response:

We thank the reviewer for the suggestion. We have added the description in the revised manuscript (track changes, line 359-362).

A total of 119 serum samples were tested using both assays. Compared to ID.vet-cELISA, the newly developed NP-cELISA demonstrated a sensitivity of 100% (66/66) and a specificity of 67.9% (36/(36+17)). The overall agreement between the two assays was 85.7% ((66+36)/119) (Table 2). To investigate the discordance between the serological results of the NP cELISA and ID.vet-cELISA, the WOAHA-recommended HI assay was used for further confirmation. The coincidence rates of NP-cELISA and ID.vet-cELISA with HI were 87.4% (77+27)/119) and 78.2% (63+30)/119), respectively (Table 3). **Among the 17 samples that were positive in the NP-cELISA but negative in the ID.vet-cELISA, 14 tested positive and 3 tested negative in the HI assay. The concordance between the NP-cELISA and HI assay results was higher than that between ID.vet-cELISA and HI assay results** (track changes, line 359-362).

Re: Spectrum00939-25R2 (Development and application of a NP-cELISA for the detection of nucleoprotein antibodies of equine influenza virus)

Dear Prof. Xiaojun Wang:

Your manuscript has been accepted, and I am forwarding it to the ASM production staff for publication. Your paper will first be checked to make sure all elements meet the technical requirements. ASM staff will contact you if anything needs to be revised before copyediting and production can begin. Otherwise, you will be notified when your proofs are ready to be viewed.

Sincerely,
Yunyu Chen
Editor
Microbiology Spectrum